# Provably Robust Explainable Graph Neural Networks against Graph Perturbation Attacks

Jiate Li[1], Meng Pang[2], Yun Dong[3], Jinyuan Jia[4], Binghui Wang[1]
[1]Illinois Institute of Technology, USA, [2]Nanchang University, China
[3]Milwaukee School of Engineering, USA [4]The Pennsylvania State University, USA

## Abstract

Explainable Graph Neural Networks (XGNNs) have garnered increasing attention for enhancing the transparency of Graph Neural Networks (GNNs), which are the leading methods for learning from graph-structured data. While existing XGNNs primarily focus on improving explanation quality, their robustness under adversarial attacks remains largely unexplored. Recent studies have shown that even minor perturbations to graph structure can significantly alter the explanation outcomes of XGNNs, posing serious risks in safety-critical applications such as drug discovery.

In this paper, we take the first step toward addressing this challenge by introducing `XGNNCert`, the first provably robust XGNN. `XGNNCert` offers formal guarantees that the explanation results will remain consistent, even under worst-case graph perturbation attacks, as long as the number of altered edges is within a bounded limit. Importantly, this robustness is achieved without compromising the original GNN's predictive performance. Evaluation results on multiple graph datasets and GNN explainers show the effectiveness of `XGNNCert`. Source code is available at `https://github.com/JetRichardLee/XGNNCert`.

## 1 Introduction

Explainable Graph Neural Network (XGNN) has emerged recently to foster the trust of using GNNs—it provides a human-understandable way to interpret the prediction by GNNs. Particularly, given a graph and a predicted node/graph label by a GNN, XGNN aims to uncover the *explanatory edges* (and the connected nodes) from the raw graph that is crucial for predicting the label (see Figure 1(a) an example). Various XGNN methods (Ying et al., 2019; Luo et al., 2020; Yuan et al., 2021; Zhang et al., 2022; Wang & Shen, 2023; Behnam & Wang, 2024) have been proposed from different perspectives (more details see Section 5) and they have also been widely adopted in applications including disease diagnosis (Pfeifer et al., 2022), drug analysis (Yang et al., 2022; Wang et al., 2023b), fake news spreader detection (Rath et al., 2021), and molecular property prediction Wu et al. (2023).

While existing works focus on enhancing the explanation performance, the robustness of XGNNs is largely unexplored. Li et al. (2024) observed that well-known XGNN methods (e.g., GNNExplainer (Ying et al., 2019), PGExplainer (Luo et al., 2020)) are vulnerable to graph perturbation attacks —Given a graph, a GNN model, and a GNN explainer, an adversary can slightly perturb a few edges such that the GNN predictions are accurate, but the explanatory edges outputted by the GNN explainer on the perturbed graph is drastically changed. This attack could cause serious issues in the safety/security-critical applications such as drug analysis. For instance, Wang et al. (2023b) designs an XGNN tool Drug-Explorer for drug repurposing (reuse existing drugs for new diseases), where users input a drug graph and the tool outputs the visualized explanation result (i.e., important chemical structure) useful for curing the diseases. If such tool is misled on adversarial purposes (i.e., adversary inputs a carefully designed perturbed graph), it may recommend invalid drugs with harmful side-effects. Therefore, it is crucial to design defenses for GNN explainers against these attacks.

Generally, defense strategies can be classified as *empirical defense* and *certified defense*. Empirical defenses often can be broken by stronger/adaptive attacks, as verified in many existing works on defending against adversarial examples (Carlini et al., 2019) and adversarial graphs (Zhang et al., 2021a; Yang et al., 2024). We notice two empirical defense methods (Bajaj et al., 2021; Wang et al., 2023c) have been proposed to robustify XGNNs against graph perturbations. Likewise, we found they

are ineffective against stronger attacks proposed in Li et al. (2024) (see Table 4). In this paper, we hence focus on designing certified defense for XGNNs against graph perturbation attacks. An XGNN is said certifiably robust against a bounded graph perturbation if, for any graph perturbation attack with a perturbation budget that does not exceed this bound, the XGNN consistently produces the same correct explanation (formal definition is in Definition 1). There are several technical challenges. First, GNN explanation and GNN classification are coupled in XGNNs. Robust GNN classifiers do not imply robust GNN explainers, and claiming robust explanations without correct GNN classification is meaningless[1]. It is thus necessary to ensure both robust GNN classification and robust GNN explanation. Second, there is a fundamental difference to guarantee the robustness of GNN classifiers and GNN explainers. This is because GNN classifiers map a graph to a label, while GNN explainers map a graph to an edge set. All existing certified defenses (Jia et al., 2020; Jin et al., 2020; Wang et al., 2021a; Xia et al., 2024; Li & Wang, 2025) against graph perturbations focus on the robustness of GNN *classifiers* and cannot be applied to robustify GNN explainers.

In this work, we propose `XGNNCert`, the first certifiably robust XGNN against graph perturbation attacks. Given a testing graph, a GNN classifier, and a GNN explainer, `XGNNCert` consists of three main steps. First, we are inspired by existing defenses for classification (Levine & Feizi, 2020b; Jia et al., 2021; 2022; Xia et al., 2024; Yang et al., 2024; Li & Wang, 2025) that divide an input (e.g., image) into multiple non-overlapping parts (e.g., patches). However, directly applying the idea to divide the test graph into multiple non-overlapping subgraphs does not work well for robustifying GNN explainers. One reason is that it is hard for the GNN explainer to determine the groundtruth explanatory edges from each subgraph due to its sparsity. To address it, we propose to leverage both the test graph and its complete graph for "hybrid" subgraph generation. An innovation design here is only a bounded number of hybrid subgraphs could be affected when the test graph is adversarially perturbed with a bounded perturbation, which is the requirement for deriving the robustness guarantee. Second, we build a majority-vote classifier on GNN predictions for the generated hybrid subgraphs, and a majority-vote explainer on GNN explanations for interpreting the prediction of the hybrid subgraphs. Last, we derive the certified robustness guarantee. Particularly, `XGNNCert` guarantees the majority-vote classifier yields the same prediction, and majority-vote explainer outputs close explanatory edges for the perturbed testing graph under arbitrary graph perturbations, when the number of perturbed edges is bounded (which we call *certified perturbation size*).

We evaluate `XGNNCert` on multiple XGNN methods on both synthetic and real-world graph dataset with groundtruth explanations. Our results show `XGNNCert` does not affect the normal explanation accuracy without attack. Moreover, `XGNNCert` shows it can guarantees at least 2 edges are from the 5 groundtruth explanatory edges, when averaged 6.2 edges are arbitrarily perturbed in testing graphs from the SG+House dataset. Our major contributions are as follows:

- We propose `XGNNCert`, to our best knowledge, the first certified defense for explainable GNN against graph perturbation attacks.
- We derive the deterministic robustness guarantee of `XGNNCert`.
- We evaluate `XGNNCert` on multiple graph datasets and GNN explainers and show its effectiveness.

## 2 BACKGROUND AND PROBLEM FORMULATION

**GNN and XGNN:** We denote a graph as $G = (\mathcal{V}, \mathcal{E})$, that consists of a set of nodes $v \in \mathcal{V}$ and edges $e_{u,v} \in \mathcal{E}$. A GNN, denoted as $f$, takes a graph $G$ as input and outputs a predicted label $y = f(G) \in \mathcal{C}$, with $\mathcal{C}$ including all possible labels. For instance, $y$ can be defined on the graph $G$ in graph classification, or on a specific node $v \in \mathcal{V}$ in node classification. An XGNN, denoted as $g$, uncovers the key component in $G$ that contributes to the GNN prediction $y$. In this paper, we focus on the widely-studied edge explanations where $g$ takes $(G, y)$ as input and determines the important edges in $G$. Particularly, this type of XGNN learns importance scores $\mathbf{m}$ for all edges $\mathcal{E}$; and selects the edges $\mathcal{E}_k \subseteq \mathcal{E}$ with the top $k$ scores in $\mathbf{m}$ as the explanatory edges, where $k$ is a hyperparameter of the XGNN. Formally, $\mathbf{m} = g(G, y), \mathcal{E}_k = \mathcal{E}.top_k(\mathbf{m}) = \mathcal{E}.top_k(g(G, y))$.

**Adversarial Attack on XGNN:** Given a graph $G$ and a prediction $y$ (by a GNN $f$), an XGNN $g$ and its explanatory edges $\mathcal{E}_k$. We notice that an attacker can perturb the graph structure to mislead the

---

[1]Our additional experiments in Figure 12 in Appendix C also validate that if the GNN classifier is deceived, the explanation result would be drastically different compared with the groundtruth explanation.

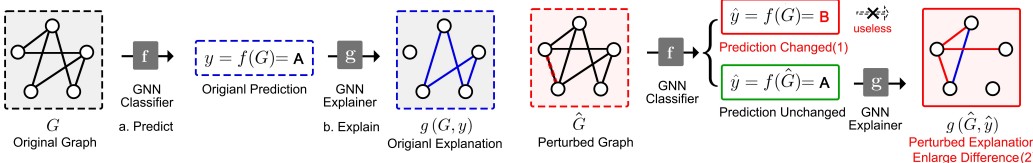

(a) GNN Explanation          (b) Adversarial Attack on GNN Explanation

Figure 1: (a) GNN for graph classification and GNN explanation. A GNN classifier $f$ first predicts a label $y$ for the graph $G$, and then a GNN explainer $g$ interprets the predicted label $y$ to produce the explanatory edges $\mathcal{E}_k$. (b) Two possible graph perturbation attacks on the GNN explainer $g$: 1) the GNN prediction $\hat{y}$ on the perturbed graph $\hat{G}$ is different from $y$; 2) the GNN prediction on $\hat{G}$ is kept, but the explanatory edges $\hat{\mathcal{E}}_k$ outputted by $g$ after the attack is largely different from $\mathcal{E}_k$.

XGNN $g$. To be specific, the attacker could delete edges from $G$ (to ensure stealthiness, the attacker does not delete edges in the explanatory edges $\mathcal{E}_k$, as otherwise it can be easily identified) or add new edge into $G$. We denote the adversarially perturbed graph as $\hat{G} = (\mathcal{V}, \hat{\mathcal{E}})$, with an attack budget $M$ (i.e., the total number of deleted and added edges is no more than $M$). On the perturbed graph $\hat{G}$, $f$ gives a new prediction $\hat{y} = f(\hat{G})$, and $g$ produces new explanatory edges $\hat{\mathcal{E}}_k = \hat{\mathcal{E}}.top_k(g(\hat{G}, \hat{y}))$.

We assume the attacker has two ways to attack an XGNN, as illustrated in Fig 1 (b).

1. *The attacker simply misleads the GNN prediction.* Note that if the prediction is changed (i.e., $\hat{y} \neq y$), even $\hat{\mathcal{E}}_k = \mathcal{E}_k$, the explanation is not useful as it explains the wrong prediction. This attack can be achieved via existing evasion attacks on GNNs, e.g., Dai et al. (2018); Zügner et al. (2018); Xu et al. (2019); Mu et al. (2021); Wang et al. (2022a).

2. *A more stealthy attack keeps the correct prediction, but largely deviates the explanation result.* That is, the attacker aims to largely enlarge the difference between $\mathcal{E}_k$ and $\hat{\mathcal{E}}_k$ with the prediction unchanged (Li et al., 2024).

**Problem Formulation:** The above results show existing XGNNs are vulnerable to effective and stealthy graph perturbation attacks. Also, as various works (Carlini et al., 2019; Mujkanovic et al., 2022) have demonstrated, empirical defenses often can be broken by advanced/stronger attacks. Such observations and past experiences motivate us to design *certifiably robust XGNNs*, i.e., that can defend against the *worst-case* graph perturbation attacks with a bounded attack budget.

**Definition 1** ($(M_\lambda, \lambda)$-Certifiably robust XGNN)**.** We say an XGNN is $(M_\lambda, \lambda)-$certifiably robust, if, for *any* graph perturbation attack with no more than $M_\lambda$ perturbed edges on a graph $G$, the GNN prediction on the induced perturbed graph $\hat{G}$ always equals to the prediction $y$ on $G$, and there are at least $\lambda$ ($\leq k$) same edges in the explanatory edges $\hat{\mathcal{E}}_k$ after the attack and the explanatory edges $\mathcal{E}_k$ without the attack. We also call $M_\lambda$ the *certified perturbation size*. Further, we denote by $M_\lambda^*$ the *maximal $M_\lambda$* associated with a $\lambda$, for which a specific XGNN remains certifiably robust.

*Remark:* A smaller $\lambda$ implies a larger $M_\lambda^*$. When $\lambda = k$, a robust XGNN should guarantee $\hat{\mathcal{E}}_k = \mathcal{E}_k$. In this paper, we focus on deriving the certifiably robust XGNN for the graph-level classification task. The adaptation of the proposed defense techniques (in Section 3) to other graph-related tasks, such as node-level classification and edge-level classification, is discussed in Appendix D.

## 3 XGNNCERT: OUR CERTIFIABLY ROBUST XGNN

In this section, we propose XGNNCert, our certifiably robust XGNN against graph perturbation attacks. Given a testing graph, a GNN classifier, and a GNN explainer, XGNNCert consists of three major steps. *1) Hybrid subgraphs generation:* it aims to generate a set of subgraphs that leverage the edges from both the testing graph and its complete graph. *2) Majority-voting based classification and explanation:* it builds a majority-vote based classifier (called voting classifier) on GNN predictions for the hybrid subgraphs, as well as a majority-vote based explainer (called voting explainer) on GNN explanations for interpreting the predicted label of the hybrid subgraphs. *3) Deriving the certified robustness guarantee:* based on the generated subgraphs, our voting classifier and voting explainer, it derives the maximum perturbed edges, such that our voting classifier guarantees the same prediction on the perturbed graph and testing graph, and our voting explainer guarantees the explanation results on the perturbed graph and the clean graph are close. Figure 2 shows an overview of our XGNNCert.

Figure 2: Overview of the proposed three-step certifiably robust XGNN.

## 3.1 HYBRID SUBGRAPHS GENERATION

A straightforward idea is to adapt the existing defense strategy for classification (Levine & Feizi, 2020b; Xiang et al., 2021; Jia et al., 2021; 2022; Xia et al., 2024) that divides an input into multiple *non-overlapping* parts. Particularly, one can divide a graph into multiple non-overlapping subgraphs, such that only a bounded number of subgraphs are affected when the graph is adversarially perturbed under a bounded perturbation. However, this strategy does not work well to robustify GNN explanation (Our results in Section 4 also validate this) due to two reasons: (1) Every edge in a graph appears only once in all subgraphs. This makes it hard for the GNN explainer to ensure the groundtruth explanatory edges to have higher scores than non-explanatory edges. (2) All subgraphs only contain existent edges in the graph, while nonexistent edges can be inserted into the graph during the attack and their importance for explanation needs to be also considered. To address the challenge, we develop a hybrid subgraph generation method, that consists of two steps shown below.

**Generating Subgraph Indexes via Hash Mapping:** We use the hash function (e.g., MD5) to generate the subgraph indexes as done in Xia et al. (2024); Yang et al. (2024)[2]. A hash function takes input as a bit string and outputs an integer (e.g., within a range $[0, 2^{128} - 1]$). Here, we propose to use the string of edge index as the input to the hash function. For instance, for an edge $e = (u, v)$, we denote its string as $\text{str}(u) + \text{str}(v)$, where the $\text{str}(\cdot)$ function transfers the index number into a string in a fixed length (filled with prefix "0"s), and "+" means the string concatenation[3]. Then we can map each edge using the hash function to a unique index. Specifically, we denote the hash function as $h$ and assume $T$ groups in total. Then for every edge $e = (u, v)$, we compute its subgraph index $i_e$ as[4].

$$i_e = h[\text{str}(u) + \text{str}(v)] \bmod T + 1. \tag{1}$$

**Generating Hybrid Subgraphs:** Based on the hash function, we can construct a set of $T$ subgraphs for any graph. However, instead of only using existent edges in the given graph to construct subgraphs, we propose to also use *nonexistent edges* to promote the robustness performance for GNN explainers. *A key requirement is: how to guarantee only a bounded number of subgraphs are affected when the original graph is adversarially perturbed.* To address it, we innovatively propose to use the *complete graph*, and our theoretical results in Theorem 2 show the requirement can be satisfied.

*Dividing the input graph into subgraphs:* For an input graph $G = (\mathcal{V}, \mathcal{E})$, we use $\mathcal{E}^i$ to denote the set of edges whose subgraph index is $i$, i.e., $\mathcal{E}^i = \{\forall e \in \mathcal{E} : i_e = i\}$. Then, we can construct $T$ subgraphs for $G$ as $\{G^i = (\mathcal{V}, \mathcal{E}^i) : i = 1, 2, \cdots, T\}$.

*Dividing the complete graph into subgraphs:* We denote the complete graph of $G$ as $G_C = (\mathcal{V}, \mathcal{E}_C), \mathcal{E}_C = \{(u, v), \forall u, v \in \mathcal{V} : u < v\}$. Similarly, we can divide $G_C$ into $T$ subgraphs using the same hash function. First, the edges having a subgraph index $i$ is denoted as $\mathcal{E}_C^i = \{\forall e \in \mathcal{E}_C : i_e = i\}$. Then, we create the $T$ subgraphs for $G_C$ as: $\{G_C^i = (\mathcal{V}, \mathcal{E}_C^i) : i = 1, 2, \cdots, T\}$.

---

[2] Our theoretical results require the graph division function has two important properties: 1) It is deterministic, such that each edge and node in a graph is deterministically mapped into only one subgraph. This property is the core to derive our theoretical results. 2) It is independent of the graph structure, as otherwise an attacker may reverse-engineer the function to find the relation between the output and input, and possibly break the defense. The used hash function can achieve both properties.

[3] For instance, with a 4-bit length, an edge 12-21 is represented as the string "0012" and "0021", respectively. Then the concatenated string between the edge 12-21 is "00120021".

[4] We put the node with a smaller index (say $u$) first and let $h[\text{str}(v) + \text{str}(u)] = h[\text{str}(u) + \text{str}(v)]$.

*Hybrid subgraphs:* Now we combine subgraphs $\{G^i\}$ with $\{G^i_C\}$ to construct the hybrid subgraphs. For each subgraph $G^i$, we propose to combine it with a fraction (say $p$) of the subgraphs in $\{G^i_C\}$ to generate a hybrid subgraph, denoted as $G^i_H$. There are many ways for the combination, and the only constraint is that the subgraph $G^i_C$ with the same subgraph index $i$ as $G^i$ is not chosen in $G^i_H$, in order to maintain the information from the original subgraph $G^i$ (otherwise it is overlaid by $G^i_C$). Let $\mathcal{T}_{-i} = \mathcal{T} \setminus i$ be the index set not including $i$. For instance, we can choose $\lfloor pT \rfloor$ indexes, denoted as $\mathcal{T}^p_{-i}$, from $\mathcal{T}_{-i}$ uniformly at random. Then a constructed hybrid subgraph is $G^i_H = (\mathcal{V}, \mathcal{E}^i_H)$, where

$$\mathcal{E}^i_H = (\cup_{j \in \mathcal{T}^p_{-i}} \mathcal{E}^j_C) \cup \mathcal{E}^i. \tag{2}$$

Note that a too small or too large $p$ would degrade the explanation performance. This is because a too large $p$ would make excessive nonexistent edges be added in each $G^i_H$, and a too small $p$ would make explanatory edges be difficult to have higher important scores than non-explanatory edges. Our results show the best performance is often achieved with a modest $p$, e.g., $p \in [0.2, 0.4]$.

With the built hybrid subgraphs, we prove in Theorem 2 that for any two graphs with $M$ different edges (but same nodes), there are at most $M$ different ones between their respective hybrid subgraphs. *We emphasize this is the key property to derive our certified robustness guarantee in Section 3.3.*

**Theorem 2** (Bounded number of different subgraphs). *For any two graphs $G = (\mathcal{V}, \mathcal{E})$, $\hat{G} = (\mathcal{V}, \hat{\mathcal{E}})$ satisfying $|\mathcal{E} \setminus \hat{\mathcal{E}}| = M$. The corresponding hybrid subgraphs generated using the above strategy are denoted as $\{G^i_H\}$ and $\{\hat{G}^i_H\}$, respectively. Then $\{G^i_H\}$ and $\{\hat{G}^i_H\}$ have at most $M$ different graphs.*

*Proof.* See Appendix A.1. □

### 3.2 MAJORITY VOTING-BASED CLASSIFICATION AND EXPLANATION

Inspired by existing works (Levine & Feizi, 2020b; Jia et al., 2022; Xia et al., 2024; Yang et al., 2024), we propose to use the majority vote to aggregate the results on the hybrid subgraphs. We then design a voting classifier and voting explainer that can respectively act as the robust GNN classifier and robust GNN explainer, as expected. Assume we have a testing graph $G$ with label $y$, a set of $T$ hybrid subgraphs $\{G^i_H\}$ built from $G$, a GNN classifier $f$, and a GNN explainer $g$.

**Voting Classifier:** We denote by $n_c$ the votes of hybrid subgraphs classified as the label $c$ by $f$, i.e.,

$$n_c = \sum_{i=1}^{T} \mathbb{I}(f(G^i_H) = c), \forall c \in \mathcal{C}, \tag{3}$$

where $\mathbb{I}(\cdot)$ is an indicator function. Then, we define our voting classifier $\bar{f}$[5] as:

$$\bar{f}(G) = \arg \max_{c \in \mathcal{C}} n_c. \tag{4}$$

**Voting Explainer:** Recall that when a GNN explainer interprets the predicted label for a graph, it first learns the importance scores for all edges in this graph and then selects the edges with the highest scores as the explanatory edges. Motivated by this, we apply $g$ on the hybrid subgraphs having the same predicted label as the majority-voted label to obtain the explanatory edges, and then vote the explanatory edges from these hybrid subgraphs. Edges with top-$k$ scores are the final explanatory edges. Specifically, for each $G^i_H$, we apply $g$ to obtain its edges' importance scores $\mathbf{m}^i = g(G^i_H, \bar{f}(G))$. We define the votes $n^\gamma_e$ of each edge $e \in \mathcal{E}_C$ as the times that its importance score $m^i_e$ is no less than $\gamma$ fraction of the largest scores in every hybrid subgraph $G^i_H$ with the prediction $f(G^i_H) = \bar{f}(G)$:

$$n^\gamma_e = \sum_{i=1}^{T} \mathbb{I}(m^i_e \geq \mathbf{m}^i_{(\gamma)}) \cdot \mathbb{I}(f(G^i_H) = \bar{f}(G)), \forall e \in \mathcal{E}_C, \tag{5}$$

where $\mathbf{x}_{(\gamma)}$ means the $\gamma \cdot \texttt{size}(\mathbf{x})$ largest element in $\mathbf{x}$ and $\gamma$ is a tuning hyperparameter (we will study its impact in our experiments). We denote $\mathbf{n}^\gamma$ as the set of votes for all edges in $\mathcal{E}_C$. Then we define our voting explainer $\bar{g}^\gamma$ as outputting the edges from $G$ with the top-$k$ scores in $\mathbf{n}^\gamma$[6]:

$$\bar{g}^\gamma(G, \bar{f}(G)) = \mathcal{E}.top_k(\mathbf{n}^\gamma). \tag{6}$$

---

[5] $\bar{f}$ returns a smaller label index when ties exist.

[6] When two edges have the same $\mathbf{n}^\gamma$, the edge with a smaller index is selected by $\bar{g}^\gamma$.

*Remark:* Traditional GNN classifiers are designed to be node permutation invariant (Kipf & Welling, 2017; Veličković et al., 2018; Xu et al., 2019), meaning that the model's predictions remain consistent regardless of how the nodes in the graph are permuted. In contrast, our voting classifier is node permutation variant due to the properties of the hash function. This implies that both the classification and explanation performances of XGNNCert may vary depending on the node ordering. However, we empirically observed that XGNNCert's performance remains relatively stable across different node permutations (see Table 9 in Appendix D). Moreover, recent studies (Loukas, 2020; Papp et al., 2021; Huang et al., 2022) suggest that node-order sensitivity can actually enhance the expressivity and generalization capabilities of GNNs. Additional discussions are provided in Appendix D.

### 3.3 Certified Robustness Guarantee

In this section, we derive the certified robustness guarantee against graph perturbation attacks using our graph division strategy and introduced robust voting classifier and voting explainer.

We first define some notations. We let $y = \bar{f}(G)$ by assuming the voting classifier $\bar{f}$ has an accurate label prediction, and $\mathcal{E}_k = \bar{g}^\gamma(G, y)$ by assuming the voting explainer $\bar{g}$ has an accurate explanation. We denote $\bar{G} = (\mathcal{V}, \bar{\mathcal{E}})$ as the complement of $G$, and $\bar{\mathcal{E}}_M$ the edges in $\bar{\mathcal{E}}$ with top-$M$ scores in $\mathbf{n}^\gamma$. We introduce the non-existent edges $\bar{\mathcal{E}}_M$ with top scores by considering that, in the worst-case attack with $M$ edge perturbations, $\bar{\mathcal{E}}_M$ would be chosen to compete with the true explanatory edges.

**Theorem 3** (Certified Perturbation Size $M_\lambda$ for a given $\lambda$). *Assume $y \in \mathcal{C}$ and $b \in \mathcal{C} \setminus \{y\}$ be the class with the highest votes $n_y$ and second highest votes by Eqn (3), respectively. Assume further the edge $l \in \mathcal{E}_k$ is with the $\lambda$-th highest votes $n_l^\gamma$, and edge $h_M \in \bar{\mathcal{E}}_M \cup (\mathcal{E} \setminus \mathcal{E}_k)$ with the $(k - \lambda + 1)$-th highest votes $n_{h_M}^\gamma$ in $\mathbf{n}^\gamma$ by Eqn (5) ($h_M$ hence depends on $M$). Then $M_\lambda$ satisfies:*

$$M_\lambda \leq M^* = \min\left(\left\lfloor \frac{n_y - n_b + \mathbb{I}(y < b) - 1}{2} \right\rfloor, M_h\right), \text{ where} \tag{7}$$

$$M_h = \max M, \quad s.t. \quad n_l^\gamma - n_{h_M}^\gamma + \mathbb{I}(l < h_M) > 2M. \tag{8}$$

*Proof.* See Appendix A.2. □

*Remark:* We have the following remarks from Theorem 3:

- Our voting classifier and voting explainer can tolerate $M^{*}$[7] perturbed edges.
- Our voting classifier can be applied for *any* GNN classifier and our voting explainer for any GNN explainer that outputs edge importance score.
- Our certified robustness guarantee is deterministic, i.e., it is true with a probability of 1.

## 4 Evaluation

### 4.1 Experimental Setup

**Datasets:** As suggested by (Agarwal et al., 2023), we choose datasets with groundtruth explanations for evaluation. We adopt the synthetic dataset "SG-Motif", where each graph has a label and "Motif" is the groundtruth explanation that can be "House", "Diamond", and "Wheel". We also adopt two real-world graph datasets (i.e., Benzene and FC) with groundtruth explanations from Agarwal et al. (2023). Their dataset statistics are described in Table 5 in Appendix C. For each dataset, we randomly sample 70% graphs for training, 10% for validation, and use the remaining 20% graphs for testing.

**GNN Explainer and Classifier:** Recent works (Funke et al., 2022; Agarwal et al., 2023) show many GNN explainers (including the well-known GNNExplainer Ying et al. (2019)) are unstable, i.e., they yield significantly different explanation results under different runs. We also validate this and show results in Table 8 in Appendix. This makes it hard to evaluate the explanation results in a consistent or predictable way. To avoid the issue, we carefully select XGNN baselines with stable explanations: PGExplainer (Luo et al., 2020), Refine (Wang et al., 2021b), and GSAT (Miao et al., 2022). We also select three well-known GNNs as the GNN classifier: GCN (Kipf & Welling, 2017), GSAGE (Hamilton et al., 2017), and GIN (Xu et al., 2019). We implement these explainers and classifiers using their publicly available source code. Appendix C details our training strategy to learn the voting explainer and voting classifier in XGNNCert.

---

[7]In general, $M^* \leq M_\lambda^*$ in Def. 1. We will leave it as future work to prove whether the derived $M^*$ is tight.

| Datasets | PGExplainer | | | | | ReFine | | | | | GSAT | | | | |
|---|---|---|---|---|---|---|---|---|---|---|---|---|---|---|---|
| | Orig. | T | | | | Orig. | T | | | | Orig. | T | | | |
| | | 30 | 50 | 70 | 90 | | 30 | 50 | 70 | 90 | | 30 | 50 | 70 | 90 |
| SG+House | 0.740 | 0.658 | 0.725 | 0.725 | 0.673 | 0.707 | 0.588 | 0.690 | 0.593 | 0.564 | 0.744 | 0.759 | 0.716 | 0.673 | 0.658 |
| SG+Diamond | 0.745 | 0.704 | 0.730 | 0.729 | 0.620 | 0.569 | 0.440 | 0.499 | 0.521 | 0.398 | 0.564 | 0.426 | 0.493 | 0.558 | 0.420 |
| SG+Wheel | 0.629 | 0.587 | 0.612 | 0.571 | 0.542 | 0.604 | 0.614 | 0.626 | 0.606 | 0.462 | 0.568 | 0.491 | 0.544 | 0.612 | 0.562 |
| Benzene | 0.552 | 0.421 | 0.497 | 0.468 | 0.429 | 0.559 | 0.463 | 0.474 | 0.512 | 0.440 | 0.552 | 0.314 | 0.430 | 0.445 | 0.398 |
| FC | 0.531 | 0.385 | 0.452 | 0.373 | 0.328 | 0.503 | 0.369 | 0.447 | 0.425 | 0.314 | 0.487 | 0.350 | 0.392 | 0.412 | 0.373 |

Table 1: Explanation accuracy on the original GNN explainers and our XGNNCert.

| Datasets | GCN | | | | | GIN | | | | | GSAGE | | | | |
|---|---|---|---|---|---|---|---|---|---|---|---|---|---|---|---|
| | Orig. | T | | | | Orig. | T | | | | Orig. | T | | | |
| | | 30 | 50 | 70 | 90 | | 30 | 50 | 70 | 90 | | 30 | 50 | 70 | 90 |
| SG+House | 0.920 | 0.895 | 0.905 | 0.905 | 0.890 | 0.945 | 0.915 | 0.915 | 0.900 | 0.905 | 0.930 | 0.900 | 0.890 | 0.895 | 0.875 |
| SG+Diamond | 0.965 | 0.935 | 0.935 | 0.935 | 0.930 | 0.975 | 0.935 | 0.955 | 0.955 | 0.955 | 0.965 | 0.940 | 0.940 | 0.940 | 0.940 |
| SG+Wheel | 0.915 | 0.905 | 0.905 | 0.900 | 0.885 | 0.930 | 0.915 | 0.905 | 0.900 | 0.895 | 0.920 | 0.910 | 0.910 | 0.895 | 0.890 |
| Benzene | 0.758 | 0.746 | 0.700 | 0.723 | 0.707 | 0.792 | 0.736 | 0.754 | 0.754 | 0.754 | 0.773 | 0.725 | 0.760 | 0.718 | 0.718 |
| FC | 0.711 | 0.674 | 0.692 | 0.692 | 0.631 | 0.800 | 0.662 | 0.714 | 0.714 | 0.703 | 0.723 | 0.692 | 0.692 | 0.692 | 0.620 |

Table 2: Prediction accuracy on the original GNN classifiers and our XGNNCert.

| Datasets | $p$ | | | | $\gamma$ | | | $h$ | | |
|---|---|---|---|---|---|---|---|---|---|---|
| | 0.0 | 0.2 | 0.3 | 0.4 | 0.2 | 0.3 | 0.4 | MD5 | SHA1 | SHA256 |
| SG+House | 0.053 | 0.695 | 0.725 | 0.710 | 0.715 | 0.725 | 0.720 | 0.725 | 0.718 | 0.710 |
| SG+Diamond | 0.045 | 0.620 | 0.729 | 0.720 | 0.712 | 0.729 | 0.718 | 0.729 | 0.729 | 0.721 |
| SG+Wheel | 0.042 | 0.511 | 0.571 | 0.508 | 0.550 | 0.571 | 0.564 | 0.571 | 0.565 | 0.562 |
| Benzene | 0.102 | 0.433 | 0.468 | 0.403 | 0.440 | 0.468 | 0.452 | 0.468 | 0.472 | 0.468 |
| FC | 0.096 | 0.353 | 0.373 | 0.288 | 0.345 | 0.373 | 0.385 | 0.373 | 0.382 | 0.390 |

Table 3: Explanation accuracy of our XGNNCert under different $p$, $\gamma$, and the hash function $h$

**Evaluation Metrics:** We adopt three metrics for evaluation. *1) Classification Accuracy*: fraction of testing graphs that are correctly classified, e.g., by our voting classifier; *2) Explanation Accuracy*: fraction of explanatory edges outputted, e.g., by our voting explainer, are in the groundtruth across all testing graphs; *3) Certified Perturbation Size $M^*$ at Certified Explanation Accuracy (or $\lambda$)*: Given a testing graph with groundtruth ($k$) explanatory edges, and a predefined $\lambda$ (or certified explanation accuracy $\lambda/k$), our theoretical result outputs (at least) $\lambda$ explanatory edges on the perturbed testing graph are from the groundtruth, where the testing graph allows arbitrary $M^*$ perturbations. $M^*$ vs $\lambda$ then reports the average $M^*$ of all testing graphs for the given $\lambda$.

**Parameter Setting:** There are several hyperparameters in our XGNNCert. Unless otherwise mentioned, we use GCN as the default GNN classifier and PGExplainer as the default GNN explainer. we use MD5 as the hash function $h$ and we set $\lambda = 3$, $p = 0.3$, $T = 70$, $\gamma = 0.3$ and $k$ as Table 5. We will also study the impact of these hyperparameters on our defense performance.

## 4.2 EVALUATION RESULTS

We first show the explanation accuracy and classification accuracy of XGNNCert under no attack, to validate it can behave similarly to the conventional GNN classifier and GNN explainer. We then show the guaranteed robustness performance of our XGNNCert against the graph perturbation attack.

### 4.2.1 EXPLANATION ACCURACY AND CLASSIFICATION ACCURACY

**XGNNCert maintains the explanation accuracy and classification accuracy on the original GNN explainers and GNN classifiers:** Table 1 shows the explanation accuracy of our XGNNCert and the original GNN explainers for reference. We can observe that XGNNCert can achieve close explanation accuracies (with a suitable number of subgraph $T$) as the original GNN explainers (which have different explanation accuracies, due to their different explanation mechanisms). This shows the potential of XGNNCert as an ensemble based XGNN. We also show the classification performance of our voting classifier in XGNNCert in Table 2 and the original GNNs classifier for reference. Similarly, we can see our voting classifier learnt based on our training strategy can reach close classification accuracy as the original GNN classifiers.

**Impact of hyperparameters in XGNNCert:** Next, we will explore the impact of important hyperparameters that could affect the performance of XGNNCert.

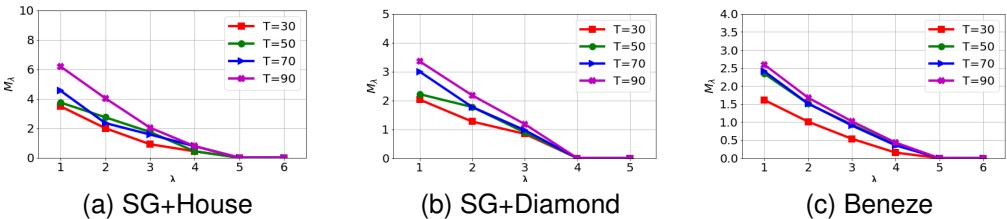

Figure 3: Certified perturbation size over all testing graphs vs. $\lambda$ on PGExplainer. The maximum $\lambda$ in x-axis equals to $k$, the number of edges in the groundtruth explanation.

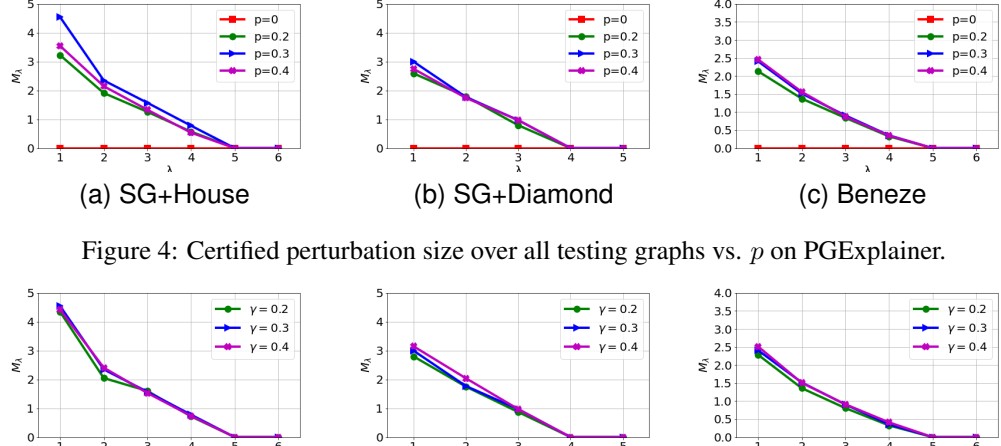

Figure 4: Certified perturbation size over all testing graphs vs. $p$ on PGExplainer.

Figure 5: Certified perturbation size over all testing graphs vs. $\gamma$ on PGExplainer.

*Impact of $T$:* Table 1 shows the explanation accuracy of `XGNNCert` with different $T$. We can see the performance depends on $T$ and the best $T$ in different datasets is different (often not the largest or smallest $T$). Note that the generated hybrid subgraphs use nonexistent edges from the complete graph. If $T$ is too small, a hybrid subgraph contains more nonexistent edges, which could exceed the tolerance of the voting explainer. In contrast, a too large $T$ yields very sparse subgraphs, making the useful information in the subgraph that can be used by the explainer be insufficient. This thus makes it hard to ensure explanatory edges have higher important scores than non-explanatory edges.

*Impact of $p$:* Table 3 shows the explanation accuracy with different $p$, the fraction of the subgraphs generated by the complete graph that are combined with the clean graphs' subgraphs. We have a similar observation that a too small or too large $p$ would degrade the explanation performance, with $p = 0.3$ obtaining the best performance overall. Note that when $p = 0$, we only use the information of the original graphs, and the explanation performance is extremely bad. That means it is almost impossible to obtain the groundtruth explanatory edges. This thus inspires us to reasonably leverage extra information not in the original graph to guide finding the groundtruth explanatory edges.

*Impact of $\gamma$:* Table 3 also shows the explanation accuracy with different $\gamma$, the fraction of the edges with the largest scores used for the voting explainer. We can observe the results are relatively stable in the range $\gamma = [0.2, 0.4]$. This is possibly due to that important edges in the original graph are mostly within these edges with the largest scores.

*Impact of $h$:* The explanation accuracy with different hash functions $h$ are shown in Table 3. We see the results are insensitive to $h$, suggesting we can simply choose the most efficient one in practice.

### 4.2.2 CERTIFIED EXPLANATION ACCURACY VS. CERTIFIED PERTURBATION SIZE

The certified robustness results are shown in Figures 3-11. Due to limited space, we only show results on three datasets and put results on the other datasets and impact of hyperparameters in Appendix C.

| Datasets | | SG+House | SG+Diamond | SG+Wheel | Benzene | FC |
|---|---|---|---|---|---|---|
| **Exp. Acc.** | **V-InfoR** | 0.693 | 0.419 | 0.439 | 0.345 | 0.217 |
| | `XGNNCert` | 0.740 | 0.729 | 0.571 | 0.468 | 0.403 |
| **Difference Fraction** | **V-InfoR** | 48.39% | 73.07% | 65.35% | 83.82% | 63.22% |
| | `XGNNCert` | 7.44% | 0.0% | 4.20% | 1.44% | 1.41% |

Table 4: Explanation accuracy and the fraction of different edges under attack in Li et al. (2024).

**Impact of** $T$**:** Figure 3 and Figures 6-8 in Appendix C show the (average) maximum certified perturbation size vs $\lambda$ with different $T$. First, XGNNCert obtains reasonable certified explanation accuracy ($\lambda/k$) against the worst-case graph perturbation, when the #perturbed edges is bounded by $M^*$. E.g., with average 6.2 edges are arbitrarily perturbed in SG+House, XGNNCert guarantees at least $\lambda = 2$ edges are from the $k = 5$ groundtruth explanatory edges. Second, there exists a tradeoff between the clean explanation accuracy and robust explanation accuracy. Specifically, as $T$ grows, the derived certified perturbation size increases in general. This means that a larger number of generated subgraphs can enlarge the gap between the largest and second-largest votes in $\mathbf{n}^\gamma$. On the other hand, the explanation accuracy (under no attack) can be decreased as shown in Section 4.2.1.

**Impact of** $p$**:** The results are in Figure 4 and Figure 9 in Appendix C. First, we observe the certified perturbation size is 0 when $p = 0$. This means, without using information in the complete graph, it is impossible to provably defend against the graph perturbation attack. Second, the certified explanation accuracies are close when $p$ is within the range $[0.2, 0.4]$ (which is different from the conclusions on explanation accuracy without attack). This implies, for each clean graph, we can use 20%-40% of the subgraphs generated by the complete graph for achieving stable certified explanation accuracy.

**Impact of** $\gamma$**:** The results are shown in Figure 5 and Figure 10 in Appendix C. Similarly, the certified results are relatively stable in the range $\gamma = [0.2, 0.4]$. The key reason could be that important edges in the original graph are mostly within the edges in these range.

**Impact of** $h$**:** The results are shown in Figure 11. Like results on explanation accuracy, we can see the hash function $h$ almost does not affect the certified explanation accuracy. Again, this suggests we can choose the most efficient one in practice.

### 4.2.3 DEFENSE EFFECTIVENESS AGAINST ADVERSARIAL ATTACK ON XGNN

We further test `XGNNCert` in the default setting against the recent adversarial attack on XGNN (Li et al., 2024), and compare with the state-of-the-art empirical defense V-InfoR (Wang et al., 2023c). We evaluate their effectiveness by allowing the attacker to change two non-explanatory edges in the graph and taking the fraction of different explanatory edges (before and after the attack) as the metric. The test results are shown in Table 4. We can observe that: Our `XGNNCert` not only achieves the theoretical defense performance and higher explanation accuracy, but also shows much better empirical defense performance than V-InfoR under the powerful attack. This is possibly due to our subgraph division and voting scheme design, which is "inherently" robust to the strongest attack—it dilutes the adversarial perturbation effect into subgraphs, and at the same time, the number of subgraphs that are affected can be bounded. In contrast, V-InfoR is an empirical defense that constrains the attack capability and is unable to defend against the strong attack.

### 4.2.4 COMPLEXITY ANALYSIS OF `XGNNCert`

Our `XGNNCert` divides each hybrid graph into $T$ subgraphs and applies a base GNN explainer to explain each subgraph. The final explanation is obtained via voting the explanation results of the $T$ subgraphs, whose computational complexity is negligible. Hence, the dominant computational complexity of `XGNNCert` is $T$ times of the base GNN explainer's. For instance, PGExplainer has a complexity of $O(S|V| + |E|)$, where $S$ is the number of optimization steps, and $|V|$ and $|E|$ are the number of nodes and edges, respectively. Therefore, `XGNNCert` with PGExplainer as the base explainer has complexity $O(TS|V| + |E|)$. Note that the explanation on $T$ subgraphs can be run in parallel, as they are independent of each other. Furthermore, each hybrid subgraph needs to store $p|V|^2$ more edges from the complete graph, where an edge is represented as a pair of node indexes in the implementation. Hence, the extra memory cost per graph is $O(pT|V|^2)$. We highlight that the extra computation and memory cost is to ensure the robustness guarantee. In other words, our `XGNNCert` obtains a robustness-efficiency tradeoff.

## 5 RELATED WORK

**Explainable GNNs:** XNNGs can be classified into *decomposition-based*, *gradient-based*, *surrogate-based*, *generation-based*, *perturbation-based*, and *causality-based* methods. *Decomposition-based methods* (Schnake et al., 2021; Feng et al., 2022) treat the GNN prediction as a score and decompose it backward layer-by-layer until reaching the input. The score of different parts of the input indicates the importance to the prediction. *Gradient-based methods* (Baldassarre & Azizpour, 2019; Pope et al., 2019) take the gradient (implies sensitivity) of the prediction wrt. the input graph, and the sensitivity is used to explain the graph for that prediction. *Surrogate-based methods* (Vu & Thai, 2020; Pereira et al., 2023) replace the complex GNN model with a simple and interpretable surrogate model. *Generation-based methods* (Lin et al., 2021; Sui et al., 2022; Shan et al., 2021; Wang et al., 2023d) use generative models to generate explanations. E.g., RCExplainer (Wang et al., 2023d) applies reinforcement learning to generate subgraphs as explanations. *Perturbation-based methods* (Ying et al., 2019; Luo et al., 2020; Wang et al., 2021b; Funke et al., 2022) uncover the important subgraph as explanations by perturbing the input graph. *Causality-based methods* (Behnam & Wang, 2024) explicitly build the structural causal model for a graph, based on the common assumption that a graph often consists of a underlying causal subgraph. It then adopts the trainable neural causal model (Xia et al., 2021) to learn the cause-effect among nodes for causal explanation.

**Adversarial attacks on GNN classifiers and explainers:** Almost all existing method focus on attacking GNN classifiers. They are classified as test-time attacks (Dai et al., 2018; Zügner et al., 2018; Ma et al., 2020; Mu et al., 2021; Wang et al., 2022a; 2023a; 2024) and training-time attacks (Xu et al., 2019; Zügner & Günnemann, 2019; Wang & Gong, 2019; Zhang et al., 2021b; Wang et al., 2023a). Test-time attacks carefully perturb test graphs so that as many as them are misclassified by a pretrained GNN classifier, while training-time attacks carefully perturb training graphs during training, such that the learnt GNN classifier mispredicts as many test graphs as possible. (Li et al., 2024) is the only method on attacking GNN explainers. It is a black-box attack (i.e., attacker has no knowledge about XGNN) that aims to corrupt GNN explanations while preserving GNN predictions.

**Certified defenses for GNN classifiers with probabilistic guarantees:** Existing certified defenses (Bojchevski et al., 2020; Wang et al., 2021a; Zhang et al., 2021b) are for GNN classifiers–they guarantee the same predicted label for a testing graph with arbitrary graph perturbation. For instance, Wang et al. (2021a) generalized randomized smoothing (Lecuyer et al., 2019; Cohen et al., 2019; Hong et al., 2022) from the continuous domain to the discrete graph domain. Zhang et al. (2021b) extended randomized ablation (Levine & Feizi, 2020c) to build provably robust graph classifiers. However, these defenses only provide probabilistic guarantees and cannot be applied to XGNNs.

**Majority voting-based certified defenses with deterministic guarantees:** This strategy has been widely used for classification models against adversarial tabular data (Hammoudeh & Lowd, 2023), adversarial 3D points (Zhang et al., 2023), adversarial patches (Levine & Feizi, 2020b; Xiang et al., 2021), adversarial graphs (Xia et al., 2024; Yang et al., 2024; Li & Wang, 2025), and data poisoning attacks (Levine & Feizi, 2020a; Jia et al., 2021; Wang et al., 2022b; Jia et al., 2022). Their key differences are creating problem-specific voters for majority voting. However, these defenses cannot be applied to robustify GNN explainers, which are drastically different from classification models.

**Certified defenses of explainable non-graph models.** A few works (Levine et al., 2019; Liu et al., 2022; Tan & Tian, 2023) propose to provably robustify explainable non-graph (image) models against adversarial perturbations. These methods mainly leverage the idea of randomized smoothing (Lecuyer et al., 2019; Cohen et al., 2019) and only provide probabilistic certificates.

## 6 CONCLUSION

We propose the first provably robust XGNN (`XGNNCert`) against graph perturbation attacks. `XGNNCert` first generates multiple hybrid subgraphs for a given graph (via hash mapping) such that only a bounded number of these subgraphs can be affected when the graph is adversarially perturbed. We then build a robust voting classifier and a robust voting explainer to aggregate the prediction and explanation results on the hybrid subgraphs. Finally, we can derive the robustness guarantee based on the built voting classifier and voting explainer against worst-case graph perturbation attacks with bounded perturbations. Experimental results on multiple datasets and GNN classifiers/explainers validate the effectiveness of our `XGNNCert`. In future work, we will enhance the certified robustness with better subgraph generation strategies and design node permutation invariant certified defenses.

ACKNOWLEDGMENTS

We thank the anonymous reviewers for their insightful reviews. This work was partially supported by the Cisco Research Award and the National Science Foundation under grant Nos. ECCS-2216926, CCF-2331302, CNS-2241713 and CNS-2339686.

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

# A  PROOFS

## A.1  PROOF OF THEOREM 2

When an edge $e$ is added to or deleted from $G$, only the subgraph $G^{i_e} = (\mathcal{V}, \mathcal{E}_{i_e})$ is corrupted after hash mapping, and all the other subgraphs $\{G^j\}_{j \neq i_e}$ are unaffected. Note that the complete graph $G_C$ is fixed and all subgraphs built from it are never affected. Then, with Equation (2), only the hybrid subgraph $G_H^{i_e}$ would be corrupted. Further, when $M$ edges from $G$ are perturbed to form $\hat{G}$, only hybrid subgraphs containing these edges would be corrupted. As some edges may be mapped to the same group index, the different subgraphs between $\{G_H^i\}$ and $\{\hat{G}_H^i\}$ is at most $M$.

## A.2  PROOF OF THEOREM 3

After the graph perturbation, we want to satisfy two requirements: (1) the voting classifier still predicts the class $y$ in the perturbed graph $\hat{G}$ with more votes than predicting any other class $\forall b \in \mathcal{C} \setminus \{y\}$; (2) the voting explainer ensures at least $\lambda$ edges in $\mathcal{E}_k$ are still in $\hat{\mathcal{E}}_k$, or at most $(k - \lambda)$ edges in $\mathcal{E}_C \setminus \mathcal{E}_k$ have higher votes than the minimum votes of the edges in $\mathcal{E}_k$.

We first achieve (1): Based on Theorem 2, with any $M$ perturbations on a graph $G$, at most $M$ hybrid subgraphs from $\{G_H^i\}$ can be corrupted. Hence, it decreases the largest votes $n_y$ at most $M$, while increasing the second-largest votes $n_b$ at most $M$ based on Eqn (3). Let $\hat{n}_y$ and $\hat{n}_b$ denote the votes of predicting the label $y$ and $b$ on the perturbed graph $\hat{G}$. We have $\hat{n}_y \geq n_y - M$ and $\hat{n}_b \leq n_b + M$. To ensure the voting classifier $\bar{f}$ still predicts $y$ for the perturbed graph $\hat{G}$, we require $\hat{n}_y > \hat{n}_b - \mathbb{I}(y < b)$ or $\hat{n}_y \geq \hat{n}_b - \mathbb{I}(y < b) + 1$, where $\mathbb{I}(y < b)$ is due to the tie breaking mechanism (i.e., we choose a label with a smaller number when ties exist). Combining these inequalities, we require $n_y - M \geq n_b + M - \mathbb{I}(y < b) + 1$, yielding

$$M \leq \lfloor \frac{n_y - n_b + \mathbb{I}(y < b) - 1}{2} \rfloor. \tag{9}$$

We now achieve (2): Recall $\bar{\mathcal{E}}_M$ the edges in $\bar{\mathcal{E}}$ with top-$M$ scores in $\mathbf{n}^\gamma$, which $\bar{\mathcal{E}}$ are edges in the complement of graph $G$. Similarly, with $M$ perturbed edges, the votes of every explanatory edge $e \in \mathcal{E}_k$ is decreased at most $M$, while the votes of every other edge $e \in \mathcal{E}_C \setminus \mathcal{E}_k$ is increased at most $M$ based on Eqn (5). Note that the edge $l \in \mathcal{E}_k$ has the $\lambda$-th highest votes $n_l^\gamma$, and the edge $h_M \in \bar{\mathcal{E}}_M \cup (\mathcal{E} \setminus \mathcal{E}_k)$ has the $(k - \lambda + 1)$-th highest votes $n_{h_M}^\gamma$. Let $\hat{n}_l^\gamma$ and $\hat{n}_{h_M}^\gamma$ denote the votes of the edge $l$ and $h_M$ on $\hat{G}$ for each $M$. Likewise, we have $\hat{n}_l^\gamma \geq n_l^\gamma - M$ and $\hat{n}_{h_M}^\gamma \leq n_{h_M}^\gamma + M$ for every $h_M$ (note $h_M$ depends on $M$). If the smallest votes $\hat{n}_l^\gamma$ of edge $l$ after the perturbation is still larger than the largest votes $\hat{n}_{h_M}^\gamma$ of the edge $h_M$, then at least $\lambda$ edges in $\mathcal{E}_k$ are still in $\hat{\mathcal{E}}_k$. This requires: $\hat{n}_l^\gamma > \hat{n}_{h_M}^\gamma - \mathbb{I}(l < h_M))$ for all $h_M$, where $\mathbb{I}(l < h_M)$ is due to the tie breaking. Combining these inequalities together, we require $n_l^\gamma - M > n_{h_M}^\gamma + M - \mathbb{I}(l < h_M), \forall M$, yielding

$$n_l^\gamma - n_{h_M}^\gamma + \mathbb{I}(l < h_M) > 2M \tag{10}$$

By satisfying both requirements, we force

$$M \leq \min \left( \lfloor \frac{n_y - n_b + \mathbb{I}(y < b) - 1}{2} \rfloor, M_h \right),$$

where $M_h = \max M, \quad s.t. \quad n_l^\gamma - n_{h_M}^\gamma + \mathbb{I}(l < h_M) > 2M$.

# B  PSEUDO CODE ON XGNNCERT

Here we provide the pseudo code of our XGNNCert, shown in Algorithm 1.

# C  EXPERIMENTAL SETUP AND MORE RESULTS

## C.1  DETAILED EXPERIMENTAL SETUP

**Dataset statistics:** Table 5 shows the statistics of the used datasets.

---

**Algorithm 1** XGNNCert: Classification, Explanation, and Certified Perturbation Size

---

**Input**: Graph $G = (\mathcal{V}, \mathcal{E})$ with $k$ explanation edges, base classifier $f$, base explainer $g$, hyperparameters: ratio $p$, ratio $\gamma$, number of subgraphs $T$, hash function $h$.

**Output**: Prediction $y$, explanation $\mathcal{E}_k$, certified perturbation size $\{M_\lambda, \lambda \in [1, k]\}$ for $G$

 1: Initialize $T$ subgraphs with empty edges $\{G^i = (\mathcal{V}, \mathcal{E}^i = \emptyset), i = 1, \cdots, T\}$.
 2: Initialize $T$ complete subgraphs with empty edges $\{G_C^i = (\mathcal{V}, \mathcal{E}_C^i = \emptyset), i = 1, \cdots, T\}$.
 3: Initialize $T$ hybrid subgraphs with empty edges $\{G_H^i = (\mathcal{V}, \mathcal{E}_H^i = \emptyset), i = 1, \cdots, T\}$.
 4: Initialize a complete edge set $\mathcal{E}_C = \{(u, v), \forall u, v \in \mathcal{V} : u < v\}$
 5: Initialize votes for all classes $\mathbf{n} = \{0\}^{|\mathcal{C}|}$, and all edges: $\mathbf{n}^\gamma = \{0\}^{|\mathcal{E}_C|}$
 6: **for** Edge $e \in \mathcal{E}_C$ **do**
 7:     Assign index $i_e = h[\text{str}(u) + \text{str}(v)] \bmod T + 1$.
 8:     **if** $e \in G$ **then**
 9:         Add $e$ into subgraph $G^{i_e}$ by $\mathcal{E}^{i_e} \cup = \{e\}$
10:     **end if**
11:     Add $e$ into complete subgraph $G_C^{i_e}$ by $\mathcal{E}_C^{i_e} \cup = \{e\}$
12: **end for**
13: **for** $i \in [1, T]$ **do**
14:     Add the $i$-th subgraph $G^i$ into $i$-th hybrid subgraph by $\mathcal{E}_H^i \cup = \mathcal{E}^i$
15:     **for** $j \in [1, i - 1] \cup [i + 1, T]$ **do**
16:         Randomlize a value $\tilde{p} \in [0, 1)$
17:         **if** $\tilde{p} \leq p$ **then**
18:             Add the $j$-th complete subgraph into $i$-th hybrid subgraph by $\mathcal{E}_H^i \cup = \mathcal{E}_C^i$
19:         **end if**
20:     **end for**
21: **end for**
22: **for** $G_C^i, i \in [1, T]$ **do**
23:     Predict $G_C^i$'s label via the base classifier: $c = f(G_C^i)$
24:     Add to the classification vote by 1: $n_c + = 1$
25: **end for**
26: Find the class with the most votes: $y = \underset{c \in \mathcal{C}}{\arg\max}\, n_c$
27: Find the class with the second most votes: $b = \underset{c \in \mathcal{C} \setminus \{y\}}{\arg\max}\, n_c$
28: Calculate the certified bound w.r.t. the classifier: $M_f = \lfloor \frac{n_y - n_b + \mathbb{I}(y < b) - 1}{2} \rfloor$.
29: **for** $G_C^i, i \in [1, T]$ **do**
30:     Explain $G_C^i$'s output via the base explainer: $\mathbf{m}^i = g(G_H^i, y)$
31:     **for** $e \in G_H^i$ **do**
32:         **if** $\mathbf{m}_e^i \geq \mathbf{m}_{(\gamma)}^i$ **then**
33:             $n_e^\gamma + = 1$
34:         **end if**
35:     **end for**
36: **end for**
37: Find the edges with top-k votes in $G$: $\mathcal{E}_k = \mathcal{E}.top_k(\mathbf{n}^\gamma)$
38: Initialize $M = 0, \{M_\lambda = 0, \lambda = 1, \cdots, k\}$
39: **while** $M_1 = M$ **do**
40:     $M + = 1$
41:     Find the edges with top-$M$ votes in $\mathcal{E}_C \setminus \mathcal{E}$: $\overline{\mathcal{E}}_M = (\mathcal{E}_C \setminus \mathcal{E}).top_M(\mathbf{n}^\gamma)$
42:     **for** $\lambda \in [1, k]$ **do**
43:         Find the edge $l \in \mathcal{E}_k$ is with the $\lambda$-th highest votes $n_l^\gamma$,
44:         Find the edge $h \in \overline{\mathcal{E}}_M \cup (\mathcal{E} \setminus \mathcal{E}_k)$ with the $(k - \lambda + 1)$-th highest votes $n_h^\gamma$ in $\mathbf{n}^\gamma$
45:         **if** $n_l^\gamma - n_h^\gamma + \mathbb{I}(l < h) > 2M$ **then**
46:             $M_\lambda = M$
47:         **end if**
48:     **end for**
49: **end while**
50: **for** $\lambda \in [1, k]$ **do**
51:     $M_\lambda = \min(M_\lambda, M_f)$
52: **end for**
53: **Return** $y, \mathcal{E}_K, \{M_\lambda, \lambda \in [1, k]\}$

---

**Hyperparameter and network architecture details in training GNN classifiers and explainers:**
We have tested the base GNN classifiers with 2 and 3 layers, the hidden neurons $\{32, 64, 128, 192\}$,

| Dataset | Graphs | $|\mathcal{V}|_{avg}$ | $|\mathcal{E}|_{avg}$ | Features | GT Graphs | GT Explanation | $|\mathcal{E}_{GT}|_{avg}$ | k |
|---|---|---|---|---|---|---|---|---|
| SG+House | 1,000 | 13.69 | 15.56 | 10 | 693 | House Shape | 6 | 6 |
| SG+Diamond | 1,000 | 10.46 | 14.03 | 10 | 486 | Diamond Shape | 5 | 5 |
| SG+Wheel | 1,000 | 12.76 | 14.07 | 10 | 589 | Wheel Shape | 8 | 8 |
| Benzene | 12,000 | 20.58 | 43.65 | 14 | 6,001 | Benzene Ring | 6 | 6 |
| Fluoride-Carbonyl (FC) | 8,6716 | 21.36 | 45.37 | 14 | 3,254 | F- and C=O | 5 | 5 |

Table 5: Datasets and their statistics.

| Ratio $p$ | SG+House | SG+Diamond | SG+Wheel | Benzene | FC |
|---|---|---|---|---|---|
| 0.0 | 0.900 | 0.925 | 0.905 | 0.723 | 0.674 |
| 0.02 | 0.895 | 0.920 | 0.900 | 0.723 | 0.692 |
| 0.03 | 0.905 | 0.935 | 0.900 | 0.723 | 0.692 |
| 0.04 | 0.895 | 0.925 | 0.900 | 0.725 | 0.662 |

Table 6: Prediction accuracy of XGNNCert with different $\rho$ (default $p = 0.3$).

| Hash function $h$ | SG+House | SG+Diamond | SG+Wheel | Benzene | FC |
|---|---|---|---|---|---|
| **MD5** | 0.905 | 0.935 | 0.900 | 0.723 | 0.692 |
| **SHA1** | 0.905 | 0.935 | 0.895 | 0.718 | 0.692 |
| **SHA256** | 0.900 | 0.935 | 0.905 | 0.725 | 0.674 |

Table 7: Prediction accuracy of XGNNCert with different hash functions. Default is "MDS".

learning rates $\{0.001, 0.002, 0.005, 0.01\}$ and training epochs $\{600, 800, 1000, 1200\}$ with the Adam optimizer (no weight decay in the training). Finally, our base GNN classifiers are all 3-layer architectures with 128 hidden neurons, the learning rate as 0.001, and the epochs as 1000.

For base GNN explainers, we simply use the configured hyperparameters in their source code. We set their hidden sizes as 64, coefficient sizes as 0.0001, coefficient entropy weights as 0.001, learning rates as 0.01, and epochs as 20. For PG Explainer, we set its first temperature as 5 and last temperature as 2. For GSAT, we set its final rate as 0.7, decay interval as 10 and decay rate as 0.1. For Refine, we set its gamma parameter as 1, beta parameter as 1 and tau parameter as 0.1.

**Training the GNN classifier and GNN explainer**: Traditionally, we only use the training graphs (with their labels) to train a GNN classifier, which is used to predict the testing graphs. In XGNNCert, however, the voting classifier uses the hybrid subgraphs (the combination of subgraphs from the testing graphs and from the corresponding complete graphs) for evaluation. To enhance the testing performance of our voting classifier, we propose to train the GNN classifier using not only the original training graphs but also the hybrid subgraphs, whose labels are same as the training graphs'[8].

Instead, the GNN explainer is directly trained on raw clean graphs due to two reasons. First, the cost of training the explainer on subgraphs is high; Second, some subgraphs do not contain groundtruth explanatory edges, making it unable to explain these subgraphs during training.

## C.2 More Results

Figure 6—Figure 8 show the certified perturbation size vs. $\lambda$ on the three GNN explainers.

Figure 9—Figure 11 show the certified perturbation size of XGNNCert vs. $p, \gamma, h$ on PGExplainer, respectively. Additionally, Table 6 and Table 7 show the prediction accuracy of XGNNCert vs. $p$ and $h$, respectively. We see the results are close, implying XGNNCert is insensitive to $p$ and $h$.

Figure 12 shows the explanation results when the GNN model is deceived. We see that explaining wrong predictions yields explanation results that are not meaningful.

---

[8]We observe the wrong prediction rate on our test hybrid subgraphs is relatively high, if we use the GNN classifier trained only on the raw training graphs. For instance, when $T = 30$, the wrong prediction rate could be range from 35% to 65% on the five datasets. This is because the training graphs used to train the GNN classifier and test hybrid subgraphs have drastically different distributions.

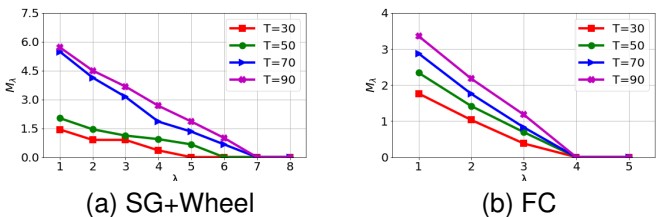

Figure 6: Certified perturbation size over all testing graphs vs. $\lambda$ on PGExplainer.

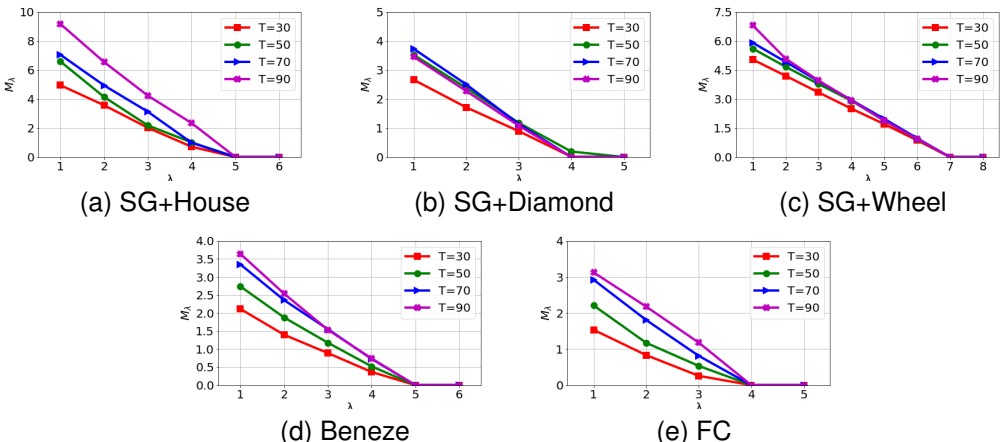

Figure 7: Certified perturbation size over all testing graphs vs. $\lambda$ on ReFine.

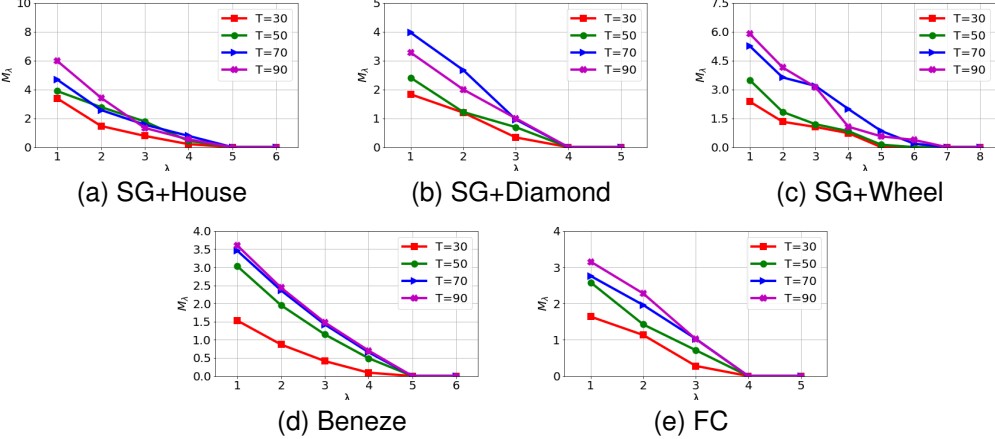

Figure 8: Maximal certified perturbation size over all testing graphs vs. $\lambda$ on GSAT.

## D DISCUSSION

**Instability of GNN explainers:** We conduct experiments on the well-known GNNExplainer (Ying et al., 2019) to show its unstable explanation results. Particularly, we run it 5 times and show the explanation results in Table 8, where "Std" is the Standard Deviation of the explanation accuracy on test data across the 5 runs, and "Change Rate" is the average fraction of different explanation edges among every pair of the 5 runs. We can see both the variance and change rate are large, meaning it is unreliable and difficult to pick any run of the result to design the robust explainer.

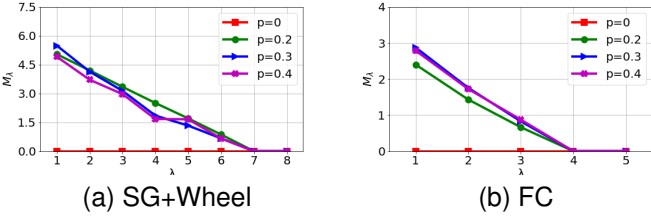

Figure 9: Certified perturbation size over all testing graphs vs. $p$ on PGExplainer.

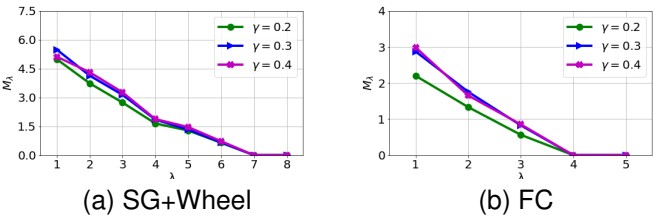

Figure 10: Certified perturbation size over all testing graphs vs. $\gamma$ on PGExplainer.

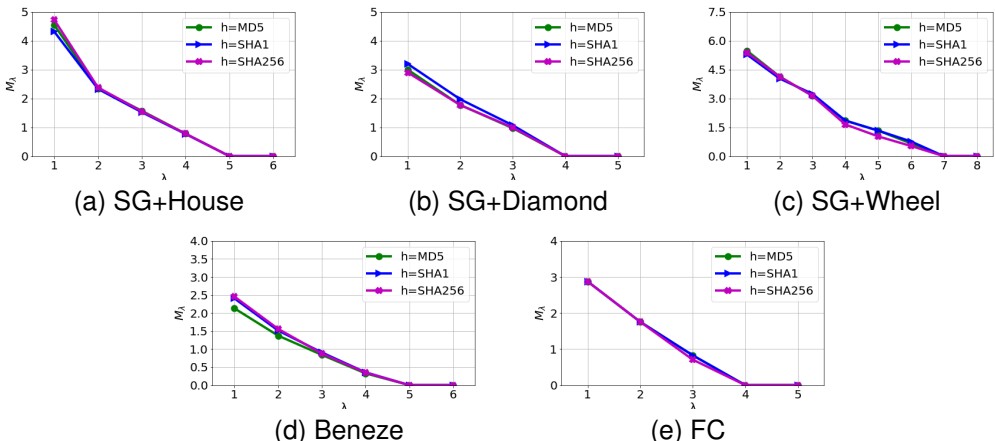

Figure 11: Certified perturbation size over all testing graphs vs. $h$ on PGExplainer.

| Dataset | SG+House | SG+Diamond | SG+Wheel | Benzene | FC |
|---|---|---|---|---|---|
| Exp. Accuracy | 0.624 | 0.368 | 0.475 | 0.276 | 0.226 |
| Std | 9.3% | 9.7% | 8.1% | 10.9% | 12.4% |
| Change Rate | 36.8% | 64.0% | 51.6% | 72.6% | 76.9% |

Table 8: Instability of GNNExplainer

| Dataset | Pred. Acc. (Avg) | Pred. Acc. (Std) | Exp. Acc. (Avg) | Exp. Acc. (Std) |
|---|---|---|---|---|
| Benzene | 0.722 | 0.002 | 0.466 | 0.007 |
| FC | 0.682 | 0.012 | 0.358 | 0.037 |

Table 9: Averaged prediction and explanation accuracy of XGNNCert on the two real-world datasets with 5 random node orderings.

**Node-order invariant vs. variant GNNs:** There exist both node-order invariant GNNs (whose outputs are insensitive to the node ordering) and node-order variant GNNs (whose outputs depend on the node ordering). Node-order invariant GNNs typically use, e.g., the mean and convolution

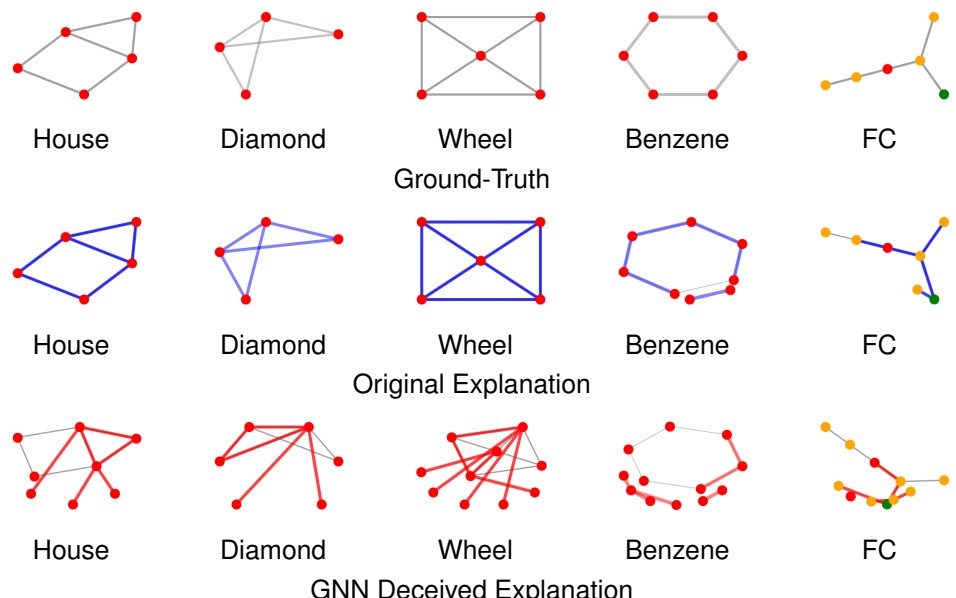

Figure 12: Examples of how an explanatory subgraph outputted by PGExplainer changes when GNN is deceived. Top Row: Groundtruth Explanation; Middle Row: Explanation under correct predictions; Bottom Row: Explanation when GNN is deceived by a graph perturbation (2 edges are perturbed).

aggregator such as GCN (Kipf & Welling, 2017), SGC (Wu et al., 2019), GIN (Xu et al., 2019), GAT (Veličković et al., 2018), GSAGE-mean (Hamilton et al., 2017)). Node-order variant GNNs are based on, e.g., random neighbor sampling (Papp et al., 2021; Rong et al., 2020; Zeng et al., 2020), LSTM aggregator (Hamilton et al., 2017), relational pooling (Murphy et al., 2019), positional embedding (Dwivedi et al., 2022; Kreuzer et al., 2021; Zhu et al., 2023).

While node-order invariant GNNs are desirable in certain cases, recent works (Loukas, 2020; Papp et al., 2021; Huang et al., 2022) show node-order variant GNNs can produce better expressivity. This ranges from the classic GSAGE with LSTM to modern graph transformers (Kreuzer et al., 2021; Zhu et al., 2023). Our GNN voting classifier is node-order variant due to the property of hash function.

To further explore the impact of node permutations on XGNNCert, we randomly permute the input graphs 5 times and report XGNNCert's average prediction and explanation accuracies on the two real-world datasets under the default setting in Table 9. We observe that XGNNCert exhibits stable prediction and explanation accuracies across the 5 runs. This demonstrates that, though XGNNCert is not inherently permutation invariant, its classification and explanation performance remain relatively stable to node permutations. We hypothesize that one possible reason for this stability is that XGNNCert augments the training graphs with a set of subgraphs to train the GNN classifier. This augmentation may mitigate the effect of node ordering, as the subgraphs are much smaller in size.

**Can this framework be extended to node-level or edge-level tasks?** Theoretically, it is possible, but needs technique adaptation. For example, in the node-level task, we are given a target node and its prediction by a GNN model, then GNN explainers aim to find the subgraph (usually from the target node's neighboring graph) that is most important for the target node's prediction. When applying the proposed framework for certifying node-level explainers, it becomes designing a graph division and voting strategy such that: with an arbitrary graph perturbation under a perturbation budget, 1) the voting classifier guarantees the correct prediction for *the target node* on the perturbed graph, and 2) the voting explainer guarantees the explanation results on the perturbed graph and clean graph are close. The current graph division strategy is not applicable as all subgraphs have disjoint nodes, while the target node should be contained in all subgraphs for the node-level task. Hence, a key challenge is how to adapt the graph division and voting strategy to satisfy 1) and 2), particularly guaranteeing only a bounded number of subgraphs is affected when predicting the target node, while the explanations of these subgraphs' predictions are also retained. We acknowledge it is interesting future work to extend the proposed framework specially for node/edge-level explanation tasks.

