# OpenReview forum: "Provably Robust Explainable Graph Neural Networks against Graph Perturbation Attacks"
_ICLR.cc/2025/Conference — ICLR 2025 Poster_

### Official Review · Reviewer_adpV · 2024-11-01

**Soundness:** 2
**Presentation:** 3
**Contribution:** 2
**Rating:** 6
**Confidence:** 3

**Summary:**

This work first studies the robustness of Explaining Graph Neural Network (XGNN) under graph perturbation attack and proposes an explainable defense mechanism. By generating hybrid subgraphs in which most of the designed voting classifier and voting explainer will give the correct prediction, this method avoids the XGNN performance degradation on both GNN's prediction and explanation under a worst-case attack. At the same time, the author provides a theoretical guarantee of the proposed defense's robustness.

**Strengths:**

1. The writing is well-structured, and the contributions are clearly stated.

2. The author provides a detailed explanation of the attack, the defense, and the theoretical guarantee.

**Weaknesses:**

1. The author claimed their method can be applied to any GNN classifier and explainer but did not show any results on different models. Only one type of classifier and explainer is used in the experiment.

2. A lot of space is spent on parameter analysis of the proposed method, which may be combined as an ablation study.

3. There is an explanation accuracy gap between XGNNCert and Baselines on datasets Benzene and FC in Table 1, where XGNNCert performs quite worse.

4. The text in Figure 2 may need to be more visible.

**Questions:**

1. The author mentions the attacker can input a perturbed graph into the drug analysis tool. How can the attacker access the data in the realistic application?

2. The author mentions that both the GNN classifier and the GNN explainer are essential for making correct explanations. Can these two models be combined into one model to defend against the graph perturbation attack rather than ensuring both model's accuracy?

3. While the author emphasizes the robustness benefits of XGNNCert, can the author show more results against different attacks? For example, shorten the content of 4.2.1 and 4.2.2 and expand 4.2.3 by showing more results on more GNN models and comparing the proposed method with more baselines.

---

> ### Author Response · Authors · 2024-11-21
> **Response by Authors**
>
> **We thank the reviewer for appreciating that the contributions are clearly stated, the paper is well-structured, and the attacks, defense, and theoretical guarantees are detailed.**
>
> **W1. Only one type of GNN classifier and GNN explainer.**
>
> This may be a misunderstanding. In Table 1-2, Figures 3-5, and Figures 6-11 in Appendix, we showed our result on three GNN classifiers (GCN, GAT and GSAGE), and three GNN explainers (PGExplainer, GSAT and Refine).
>
> **W2. parameter analysis combined as an ablation study**
>
> Sections 4.2.1 and 4.2.2 examine different aspects of our XGNNCert, making it challenging to combine their analyses. Specifically, Section 4.2.1 focuses on the normal classification and explanation accuracy of XGNNCert, while Section 4.2.2 investigates its certified explanation accuracy.
>
> **W3. XGNNCert shows a large explanation gap on Benzene with GSAT**
>
> We highlight that the normal explanation accuracy under no attack is often larger than the robust explanation accuracy under the attack on GNN explainers. Especially, our XGNNCert defends against the *worst-case attack* and it shows promising performance in most of the settings, except on Benzene and FC with GSAT. The reason may be due to Benzene and FC being the difficult datasets for explanation (note all the three explainers have relatively low performance on them), which largely affects XGNNCert’s predictions and hence explanations.
>
> **W4. Text in Figure 2 be more visible**
>
> Thanks for the suggestion! See Figure 2 in the revised version.
>
> **Q1. How can an attacker access the data in realistic applications?**
>
> There are various scenarios in which an attacker can gain access to the data. For example, one scenario involves an insider attacker with full access to the data, while another scenario involves an attacker uploading their own data (e.g., a graph) to a system, such as the Drug Explorer system mentioned in the introduction.
>
> **Q2. Can GNN classifier and explainer be combined into one model to defend against the attack**
>
> Indeed, our XGNNCert integrates the GNN classifier and GNN explainer to defend against graph perturbation attacks, as illustrated in the overview Figure 2.
>
>
> **Q3. XGNNCert against more empirical attacks.**
>
> We test the most recent attack (Li et al. ICML’24) on XGNN and adopt its powerful induction-based attack. We allow an attacker to perturb two edges, and take the *explanation edge change rate* (i.e., fraction of explanation edges not in the groundtruth after the attack) as the metric to evaluate the defense effectiveness. A *smaller value implies better defense performance*. We use the PGExplainer as the baseline GNN explainer. In our method, we set T=70 and p=0.3. The results are shown below, where our XGNNCert shows promising defense performance against the attack.
>
> |Dataset |SG-House | SG-Diamond |  SGWheel  |  Benzene | FC|
> |:------------|--------------:|:-------------:|:-------------:|:-------------:|:-------------:|
> |XGNNCert | 7.44% |  0.0% | 4.20% | 1.44% | 1.41% |

---

> > ### Comment · Reviewer_adpV · 2024-11-22
> >
> > W3. It seems like Table 1 is just about the explanation accuracy on clean graphs rather than attacked graphs which the author mentioned. Anyway, has the author tried to tune the hyper-parameters to improve these accuracies? Or it's the default flaw of XGNNCert.
> >
> > Q3. Can the author compare the performance of XGNNCert with other baselines under the attack like V-InfoR mentioned in the paper?

---

> > > ### Author Response · Authors · 2024-11-23
> > > **Response by Authors**
> > >
> > > **Thanks for the further comments!**
> > >
> > > **Response to W3:** Yes, we have tested various hyperparameter settings of XGNNCert in Table 3, but it is still not as good as the original GNN explainer (due to the inherent robustness-accuracy tradeoff). We emphasize this is the first work to provide formal robustness guarantee of XGNNs against graph perturbation attacks. We also admit it is important future work to further tighten the tradeoff (see Conclusion in the revised version).
> > >
> > > **Response to Q3:** Unfortunately, as far as we know, there are only two other works focused on XGNN (empirical) robustness: one is the *most recent* V-InfoR (NeurIPS’23) and the other is [1]. However, the code link in [1] is expired, and we are only able to show compared results with V-InfoR (and ours performs much better). If you have other baselines on XGNN robustness, please let us know and we will be happy to test them.
> > >
> > > [1] Bajaj, Mohit, et al. "Robust counterfactual explanations on graph neural networks." NeurIPS 2021.

---

> > > > ### Author Response · Authors · 2024-11-24
> > > > **Official Comment by Authors**
> > > >
> > > > Dear Reviewer adpV,
> > > >
> > > > Thank you for your time and effort on reviewing this paper and your constructive comments. We truly appreciate how your feedback has strengthened our paper.
> > > >
> > > > As the interaction period nears its conclusion, we would be grateful if you could confirm whether our new response meets your expectations or if more clarifications are needed within the limited time.
> > > >
> > > > Thank you once again for your valuable input.
> > > >
> > > > Best,
> > > >
> > > > Authors

---

> > > > ### Comment · Reviewer_adpV · 2024-11-24
> > > >
> > > > Thanks for the author's responses. I have no other questions for now. I have increased my score from 5 to 6.

---

> > > > > ### Author Response · Authors · 2024-11-24
> > > > > **Response by Authors**
> > > > >
> > > > > We’re glad that all your comments have been addressed and sincerely thank you for raising the score!

---

### Official Review · Reviewer_Z8ry · 2024-11-02

**Soundness:** 3
**Presentation:** 3
**Contribution:** 2
**Rating:** 5
**Confidence:** 3

**Summary:**

This paper proposes a certified defense method to mitigate the risk of an attacker who aims to mislead the GNN explanation. In order to achieve this goal, the calculate the maximum perturbation size to perturb in order to alter the prediction (explanation) of a graph.

**Strengths:**

1. The paper is well written.
2. The authors provide a reasonable motivation for the study (such as giving an motivating example in the introduction).
3. The proposed method empirically efficacy to obtain certified robustness and it would not hurt the original performance.
4. The method is potential to be applied for various GNN models.

**Weaknesses:**

Although the authors give a good motivation example (in the introduction section), I am still not fully convinced by the importance of the proposed study. For example, if one GNN builder aims to explain or understand their GNN model's prediction via GNN explainers, will there still be a high risk that the graph data is perturbed by an attacker? An perturbation operation could possibly easily identified by the GNN builder.

Besides, I am also wondering the tightness of the proposed certified robustness. In Figure 5, the authors only provide the calculated perturbation bounds. How close are them to the real minimum perturbation budget?

**Questions:**

Plz see the weakness.

---

> ### Author Response · Authors · 2024-11-21
> **Response by Authors**
>
> **We thank the reviewer for appreciating that the paper: 1) is well-written; 2) has a reasonable motivation to the study; and 3) provides certified robustness guarantees**
>
> **1. Will there be a high risk that the graph data is perturbed by an attacker. A GNN builder could easily identify a perturbation operation.**
>
> If a GNN builder locally and privately builds its own GNN classifier or explainer with its own private and guaranteed-to-be-clean data, then there will not be a risk. However, this is not the case in various real-world scenarios, where *the data, the GNN classifier, and the GNN explainer are from different parties*. As the example shown in the paper, an user can upload a drug graph to the Drug Explorer system for drug repurposing (Wang et a, 2023b).
>
> Under those scenarios, it is challenging for the GNN builder to identify the perturbation. In fact, various existing studies [See “Adversarial Attack on GNN Classifiers and Explainers” in Section 5] show an attacker can achieve its goal with an unnoticeably perturbation. For instance, [Zügner et al., 2018, Li et al. 2024] show an attacker can fool a GNN classifier or explainer by perturbing a graph such that the structural similarity between the perturbed graph and the clean graph is more than 99%. Also, all perturbation detection based methods fail as shown in, e.g., [a].
>
> [a] Mujkanovic et al,  Are defenses for graph neural networks robust. In NeurIPS 2022
>
>
> **2. Tightness of the derived bound.**
>
> We appreciate the reviewer's insightful question regarding the tightness of our bound. This is indeed an important aspect to consider in understanding the theoretical implications of our work. At present, we do not have formal evidence to confirm whether the bound is tight or not. This is a challenging problem as we need to consider both GNN classification and GNN explanation in the theoretical proof. We acknowledge that determining the exact tightness of the bound is an interesting and challenging open question that we aim to explore in future research.

---

> > ### Author Response · Authors · 2024-11-24
> > **Official Comment by Authors**
> >
> > Dear Reviewer,
> >
> > Thank you again for your constructive comments. We truly appreciate how your feedback has strengthened our paper.
> >
> > As the interaction period nears its conclusion, we would be grateful if you could confirm whether our responses meet your expectations or if more clarifications are needed within the limited time.
> >
> > Thank you once again for your valuable input.
> >
> > Best,
> >
> > Authors

---

> ### Author Response · Authors · 2024-12-01
>
> Dear Reviewer Z8ry,
>
> We thank you again for your appreciation on the paper's strengths and your constructive comments on the problem setting and the tightness concern.  As the discussion period comes to the close, we would be grateful if you could confirm whether our responses address your concerns or if more clarifications are needed within the limited time. We believe your further feedback would help us to improve our paper.
>
> Thank you once again for your precious input.
>
> Best,
>
> Authors

---

### Official Review · Reviewer_DXMv · 2024-11-03

**Soundness:** 3
**Presentation:** 3
**Contribution:** 3
**Rating:** 8
**Confidence:** 4

**Summary:**

The paper introduces a new architecture for explainable graph learning that allows to additionally derive a certificate regarding the set of explanations (edges) w.r.t. edge perturbations. This is achieved by splitting the graphs into subgraphs and derive an ensemble-based explainer. In particular, the well-known robustness certification strategy for majority voting classifiers is extended to the set case. The effectiveness of the method is demonstrated on several graph classification datasets.

**Strengths:**

* In general, the paper is well-written and nice to follow. Next to the well-written main parts, I want to additionally highlight the very good and thorough overview of related work and contextualization of this work in it given in Section 5.
* With a certifiably robust XGNN the paper seems to tackle a relevant problem given the brittleness of XGNNs to adversarial attacks.
* Extending robustness certification (especially w.r.t. ensembles) to explainable sets is non-trivial, conceptually, as well as w.r.t. the proof strategy.
* Through appropriate experiments, the effectiveness of the method to provide a certain robustness to structure changes is shown. Furthermore, the experiments nicely show the tradeoffs between classic XGNN methods and XGNNCert.

**Weaknesses:**

1. Currently, the experiments are not reproducible as experimental details are missing about $(i)$ the training strategy (even though Line 312 links to Appendix B for this, Appendix B does not include any experimental details), and $(ii)$ architectural details and hyperparameters of the used base classification and explainer GNNs. This is made even more critical as $(1)$ there is no code submitted for the review or indication of releasing code upon acceptance, and  $(2)$ no reproducibility statement provided.
2. The $(M_\lambda, \lambda)$ robustness definition together with the remark in Line 148 is currently confusing. In particular, for any $\lambda$, there can be many associated $M_\lambda$ such that the network is still $(M_\lambda, \lambda)$ robust. Thus, $M_\lambda$ is not uniquely indexed by $\lambda$. However, by indexing $M$ with $\lambda$, stating that $M_\lambda$ is the certified perturbation size, and explicitly remarking it depends on $\lambda$, it gives the impression as if it should be a unique function of $\lambda$ and in the first read, it was unclear to me if the authors therefore actually mean the maximal $M_\lambda$ for a particular $\lambda$. I think the confusion could be resolved by removing the index $\lambda$ (it is evident that $\lambda$ and $M$ have a certain connection and tradeoff) and optimally, also introduce the concept of a maximal $M$ after the definition with a different letter (e.g., $M^*$ as used in Thm. 3, or now $M_\lambda$, as the the maximal $M$ is uniquely determined by a given $\lambda$).
3. The goal of the work stated in lines 141-144 does not align with the contribution of the paper. In particular, the paper designs a concrete XGNN architecture based on ensembles to be able to (for the first time) derive robustness certificates. However, Eq. 1 defines a very different goal, namely optimizing over function spaces defining the XGNN and GNN classifiers, to find those functions leading to the highest certified robustness. However, the optimization over function spaces to maximize certified robustness is never mentioned or tackled in the subsequent paper. My recommendation is to remove Eq. 1, as the problem tackled by the paper is already very well motivated and formulated before providing this very different goal description. Otherwise, questions such as "what function spaces do we optimize over", or "do we have convergence guarantees", or similar, need to be asked and answered. As a third alternative, one could not state it as the goal of the current work, but a general goal the whole field/literature tries to answer/converge to.
4. Generating subgraph indexes via the proposed hash mapping has first been proposed and done by Xia et al. (2024), I would expect to make clear somewhere that their hashing is used, potentially just by stating this and citing Xia et al. (2024) in the beginning of the second paragraph in Section 3.1 (e.g., "We use the hash function (e.g., MD5) to generate the subraph indexes as done in Xia et al. (2024)."). Furthermore, regarding majority voting-based certified defenses, the work of Wang et al. (2022) already divides the input data into non-overlapping parts (in contrast to what the paper mentions in lines 526/527). Even though this work is qualitatively different, I expect the work to be cited and the statement corrected when this is explicitly discussed in lines 526/527.
5. The method is not permutation invariant. The authors discuss this in Appendix C, which however, is not linked in the main draft (please link to it). GNNs are usually developed with the goal to be permutation invariant (Hamilton 2020), and to me, it feels odd if an XGNN explanation is dependent on the concrete node order. In particular, node-order sensitivity can improve expressivity (and generalization) in the face of non-discriminatory node features (Huang et al. 2022), but I do not see that this property is particularly relevant for the developed XGNNCert as all used datasets seem to have discriminatory node features (nor any argument given that the concrete usage of permutation sensitivity improves expressivity). Addressing the permutation invariance is not critical for me to recommend acceptance, but I would like to at least see an experiment showing how brittle or stable the performance (classification and explanation-wise) of XGNNCert is to different permutations of the adjacency matrix.
6. From the sentence at line 93-94, I first had the impression that the method can be applied to node classification tasks. However, Appendix C (again not linked) states that this is not possible without adaptation. I recommend to make this clear in the main draft when introducing the background or method and link to the discussion in Appendix C.
7. In Section 3.3 the authors mention multiple times a statement similar to the one found in Line 276: "$\bar{\mathcal{E}}_M$ [are] the edges in $\bar{\mathcal{E}}$ with top-$M$ scores in $n^\gamma$". Until reading and understanding the proof it was unclear to me if such a statement means either $(i)$ the edges in $\bar{\mathcal{E}}_M$ that are among the top-$M$ edges in $n^\gamma$ given all edges (i.e., $|\bar{\mathcal{E}}_M|\le M$); or $(ii)$ $\bar{\mathcal{E}}_M$ are the top-$M$ scored edges in $\bar{\mathcal{E}}$, i.e. $|\bar{\mathcal{E}}_M|=M$. Making this clear before providing Thm. 3 would greatly improve readability of Section 3.3 and the Thm. 3 in particular.
8. Figure 4 and 5 feel repetitive to Figure 3. Maybe, it could be more interesting to show the impact of $p$ and $\gamma$ also in the main draft (i.e., Figures similar to Fig. 9 & 10). Also note that in a first read, I was overwhelmed by Line 431 linking to Figures 3-11 all at once and it was not clear to me, if I should look at that figures now and how to find all the in-between figures.

I think the paper has the potential to be a good contribution to the ICLR community. If the reproducibility weakness is addressed, I will raise my rating to a borderline score. If additionally my other weaknesses and questions are appropriately addressed, I am considering changing my score to accept (8).

**Questions:**

1. How do the hyperparameters $p$ and $h$ effect prediction accuracy? I really like Tables 1 & 2 and think having a similar investigation for $p$ and $h$ in the Appendix would be interesting.
2. Is certification possible if an edge in the ground truth explanation set is perturbed? In particular, I'm not very convinced of the stealthiness argument in line 102, as a defender would not have access to the ground truth graph, but only the perturbed version (and of course, does not have access to the ground truth explanatory edges, as this would make the method otherwise obsolete). My thinking would be yes, as from a defender perspective the $\mathcal{E}_k$ are already derived from the perturbed graph and the $\lambda$ guarantee holding for any graph reachable through $M^*$ edge changes.
3. Where does the -1 in the robustness guarantee come from? Shouldn't from the proof $M \le \lfloor \frac{n_y-n_b+\mathbb{I}(y<b)}{2}\rfloor$ suffice? What am I missing?
4. Regarding the experiments performed in Table 6, what is the instability of XGNNCert under the same experimental setup?
5. In the related work, can the authors include one or two sentences about certification of explainable AI in a non-graph context? This would be interesting to the reader, as one could ask the question of what approaches are already out there and if these are directly applicable to the graph domain.

Minor
* Figure 1: "Origianl Prediction" -> "Original Prediction" (2x)
* Line 12: "learn graph data" -> "learn on graph data"
* Line 13: "its robustness" -> "their robustness"
* Line 18: "XGNN" -> "XGNNs"
* Line 53: It could help to first define a certified defense before using the term.
* Line 265: Writing $|x|$ to denote the number of elements in $x$ is confusing, as it is not written or indicated that $x$ is a set.
* Line 278/279: "with" -> "within"
* Line 354: $M^*$@$\lambda$ <- is this intended?
* Line 363/364: "GNN explaniner" -> "GNN explainer"


**References**
Hamilton, "Graph Representation Learning", McGill University 2020
Huang et al., "Going Deeper into Permutation-Sensitive Graph Neural Networks", ICML 2022
Wang et al., "Improved Certified Defenses against Data Poisoning with (Deterministic) Finite Aggregation", ICML 2022
Xia et al., "GNNCert: Deterministic Certification of Graph Neural Networks against Adversarial Perturbations", ICLR 2024

---

> ### Author Response · Authors · 2024-11-21
> **Response by Authors - Part I**
>
> **We sincerely thank the reviewer for recognizing that the paper is well-written, the problem is novel, and the robustness certification is non-trivial. We also greatly appreciate the constructive and insightful comments.**
>
> **W1. Experimental details on (i) training strategy; (ii) network architecture; and (iii) code link for reproducibility.**
>
> (i) and (ii): Please see Appendix C.1 in the revised submission.
>
> (iii)  Please see the anonymous code link:  https://anonymous.4open.science/r/XGNNCert-0B1F
>
> **W2. Relationship between $M_{\lambda}$ and $\lambda$; and $M^\star$.**
>
> Sorry for the confusion! We guess the confusion may arise from two aspects:
>
> - We use $M_\lambda$ to denote both the certified perturbation size and the *maximal* certified perturbation size (as shown on y-axis in Figures 3-11);
>
> - We use the same notation $M^*$ without explicitly indicating its dependence on $\lambda$, although it is actually $\lambda$-dependent. Below, we provide more clarifications.
>
> In our setup and description, $M_{\lambda}$ depends on $\lambda$. To be specific, given a graph $G$ with $k$ groundtruth explanatory edges and a $\lambda$, we aim to ensure that the groundtruth explanatory edges and the outputted explanatory edges from a robust XGNN have at least $\lambda$ overlapped edges, under any attack with $M_{\lambda}$ perturbed edges (referred to as the certified perturbation size in the submission). Clearly, for different $\lambda$ values, the allowed number of perturbed edges could be different. Thus, we introduce $M_{\lambda}$ to correspond to a specific $\lambda$. Our goal is to determine the *maximal* $M_{\lambda}$ for each given $\lambda$.
>
> Additionally, we note that $M^*$ in Theorem 3 is also $\lambda$-dependent. That is, for each $\lambda$, we re-run Eqns (7)-(8) to compute $M^*$.
>
> To address this confusion, in the revised version, we will consistently use $M_{\lambda}$ to indicate the *maximal* allowed number of perturbed edges (which we still call certified perturbation size) for a specific $\lambda$.
>
> *We also included pseudo code for XGNNCert (see Algorithm 1 in Appendix B) to enhance clarity and understanding.*
>
> **W3: Remove Eq.1 as the problem tackled by the paper is already very well motivated and formulated before providing this very different goal description.**
>
> Good suggestion! We have removed Eq.1 in the revised version.
>
> **W4. 1) Make clear the hashing is used in Xia et al. (2024); 2) Cite Wang et al. (2022) and correct the statement**
>
> Thanks for the suggestion!
>
> 1) We revised the texts as suggested. Please see Line 186 in Section 3.1.
>
> 2) Please refer to Lines 521–524 in Related Work. We acknowledge that there was a mistake in our previous statement, which has now been corrected in the revised version.
>
>
> **W5. 1) Link discussion on permutation invariant vs variant in Appendix C in the main draft; 2) Show XGNNCert’s classification / explanation performance to different node permutations.**
>
> 1) See Lines 270-279 the discussion and link to Appendix D.
>
> 2) We show the results below under the default setting on the two real-world datasets (Table 9 in the revised version) where we randomly permute the input graph 5 times and report XGNNCert’s averaged classification and performance across the 5 times.
>
> | Dataset  | Pred. Acc. (Avg) | Pred. Acc. (Std) | Exp. Acc. (Avg) | Exp. Acc. (Std) |
> |----------|------------------|------------------|-----------------|-----------------|
> | Benzene  | 0.722            | 0.002            | 0.466           | 0.007           |
> | FC       | 0.682            | 0.012            | 0.358           | 0.037           |
>
>
> Though XGNNCert is not inherently permutation invariant, we see its classification and explanation performance remain relatively stable to node permutations. (Also see more explanations in Line 1059-1066 in Appendix).
>
> **W6: Linking the discussion of generalizing XGNNCert on node-classification tasks in Appendix C to the main body (e.g., background or method)**
>
> Please see Lines 145-148 in Section 2 Background in the revised version.
>
>
> **W7: Is $|\bar{\mathcal{E}}_{M}| \leq M$ or $= M$?**
>
> It is equal to $M$. We use the exact $M$ (non-existent) edges with top-M scores for our certification by considering that the attacker is allowed to perturb $M$ edges. In the worst-case attack, the M edges $\bar{\mathcal{E}}_{M}$ would be chosen to compete with the true explanatory edges. We also clarify this in the revised version (line 288-289).
>
> **W8. Show the figures on the impact of $\rho$ and $\gamma$ in the main paper**
>
> Thanks for your suggestions! See the new Figures 4-5 in the revised version.

---

> > ### Author Response · Authors · 2024-11-21
> > **Response by Authors - Part II**
> >
> > **Q1. How p and h influence the prediction?**
> >
> > See below the prediction results with different $h$ and $p$ under the default setting. We see the results are close, implying $p$ and $h$ marginally influence the prediction performance.  We also include these results in Tables 6 and 7 in the revised version.
> >
> > | $h$ | SG+House | SG+Diamond | SG+Wheel | Benzene | FC    |
> > |---------------|----------|------------|----------|---------|-------|
> > | **MD5**       | 0.905    | 0.935      | 0.900    | 0.723   | 0.692 |
> > | **SHA1**      | 0.905    | 0.935      | 0.895    | 0.718   | 0.692 |
> > | **SHA256**    | 0.900    | 0.935      | 0.905    | 0.725   | 0.674 |
> >
> > | $p$ | SG+House | SG+Diamond | SG+Wheel | Benzene | FC    |
> > |-------|----------|------------|----------|---------|-------|
> > | 0.0   | 0.900    | 0.925      | 0.905    | 0.723   | 0.674 |
> > | 0.02  | 0.895    | 0.920      | 0.900    | 0.723   | 0.692 |
> > | 0.03  | 0.905    | 0.935      | 0.900    | 0.723   | 0.692 |
> > | 0.04  | 0.895    | 0.925      | 0.900    | 0.725   | 0.662 |
> >
> >
> > **Q2: 1) Stealthiness argument on not deleting explanation edges. 2) Is it still certification possible if explanation edges are allowed to be perturbed?**
> >
> > 1) In many real-world graph datasets, particularly molecules used for GNN explanation, the groundtruth explanatory edges often exhibit inherent structures that can be identified when perturbed. For example, in the MUTAG dataset, the groundtruth includes the NO2 group (O=N=O), while in the Benzene dataset, it is the benzene ring structure. If a molecular graph lacks an NO2​ group or a benzene ring, it becomes evident that the graph has been perturbed. Therefore, we consider the setting where the attacker does not delete groundtruth explanatory edges. Notably, this setting aligns with the recent adversarial attack on XGNN (Li et al., ICML 24).
> >
> > 2) Yes, we can still derive the robustness certificate when explanatory edges are perturbed. However, the certified perturbation size would be relatively small, as the attacker could perform a trivial yet  potentially optimal attack via simply deleting the explanatory edges. It is important to note that once explanatory edges are deleted from the graph, no GNN explainer would be able to output those explanatory edges.
> >
> >
> > **Q3. Why add -1?**
> >
> > We carefully checked our proof as well as the proof in [Xia et al. 2024] and found the correct form should be $+ \mathbb{I}(y<b) -1$ not $+ \mathbb{I}(y<b) $ in the numerator.
> >
> > **Q4. The instability of XGNNCert under the same setup as in Table 6.**
> >
> > Table 6 is to explain why we do not use the non-deterministic GNN explainers (e.g., the well-known GNNExplainer) in our XGNNCert — these explainers generate relatively *unstable* results in different runs. Therefore, our XGNNCert chooses the deterministic PG-Explainer, GSAT and Refine. As XGNNCert is based on a wholly deterministic explainer, its result does NOT change with different runs (on the same input graphs).
> >
> > **Q5. Add a few sentences about certification of XAI in non-graph domains.**
> >
> > See Line 526-529 in Related Work.
> >
> > Minors: Thanks for pointing out. We fixed them in the revised version.
> >
> > **We hope our response has addressed all your comments and are happy to provide further clarifications if needed.**

---

> ### Author Response · Authors · 2024-11-24
> **Official Comment by Authors**
>
> Dear Reviewer,
>
> Thank you again for your constructive and insightful comments. We have tried our best effort to address all the concerns you raised in your review, and we truly appreciate how your feedback has significantly strengthened our paper.
>
> As the interaction period nears its conclusion, we would greatly value your response. If you have any further concerns or require additional clarifications, please do not hesitate to let us know. We would be happy to address them promptly.
>
> Best regards,
>
> Authors

---

> ### Comment · Reviewer_DXMv · 2024-11-24
>
> I thank the authors for their response and clarifications. Most of my concerns are well addressed, except for one important point I will detail below. I increase my score to accept (8) under the assumption that this point will be further addressed.
>
> The $(M_\lambda, \lambda)$-robustness definition is still unclear. Changing the first sentence to *"We say an XGNN is $(M_\lambda, \lambda)$−certifiably robust, if, for any graph perturbation attack with **a maximal number of** $M_\lambda$ perturbed edges on a graph $G$ [...]"* does not address this issue, as it did not change the semantic meaning of the definition. The authors admitted to using $M_\lambda$ to denote both, the certified perturbation size (of which there can be many for one $\lambda$) and the maximal certified perturbation size (which is unique). This is one of the main reasons I find the current definition confusing and I do think it is not helping understanding to overload such a critical symbol (and definition) for this work.
>
> I think this can be resolved by e.g. introducing an explicit symbol $M_\lambda^*$ (or similar) to denote the maximal certified perturbation size directly below the $(M_\lambda, \lambda)$-robustness definition by adding a sentence similar in meaning to:
>
> "With $M_\lambda^*$ we denote the maximal $M_\lambda$ associated to a $\lambda$ for which a given XGNN is still certifiable robust."
>
> and make clear in the following text, when what is used. This is one suggestion and I am open to other ideas by the authors.

---

> > ### Author Response · Authors · 2024-11-24
> > **Response by Authors**
> >
> > Dear Reviewer DXMv,
> >
> > We are pleased to hear that our rebuttal has addressed all but one of your comments, and we sincerely thank you for raising the score from 3 to 8.
> >
> > We greatly appreciate your suggestion to introduce the new notation $M_\lambda^*$, which indeed improves the clarity of Definition 1. We have revised the relevant sections in both the main body and the Appendix, highlighting the changes in blue in the updated version.
> >
> > Please let us know if these revisions meet your expectations or if further adjustments are needed.
> >
> > Best regards,
> >
> > Authors

---

> > > ### Comment · Reviewer_DXMv · 2024-11-26
> > >
> > > I appreciate the inclusion of $M_\lambda^*$ and think Def. 1 is now made clear. However, the further adaptions in the manuscript by the authors led to a new inconsistency. To repeat, $M_\lambda^*$ is defined as the maximal $M_\lambda$ for which a given XGNN is still certifiably robust as I proposed. However, the updated version of Thm. 3 by the authors seems to me incorrect now. In particular, Thm. 3 provides an incomplete certificate, i.e. it provides a provable lower bound to the robust accuracy of the introduced XGNN but not necessarily its exact robustness (this is inherent in the used proof strategy for ensemble-based classifiers). Thus, while it returns a certain $(M_\lambda,\lambda)$-robustness, we do not know if the returned $M_\lambda$ truly is $M_\lambda^*$ (and there will be cases where it certainly is not). Further, there may be more tight certification strategies in the future that can provide higher $M_\lambda$ than provided by Thm. 3.
> > >
> > > One way to correct this new inconsistency would be to change Thm. 3 back to the formulation provided in the original submission. Indeed, there you introduce $M^*$ representing the maximal certified perturbation size calculable by the provided certification method (which may be different to $M_\lambda^*$ as argued above). This could be combined with a remark that $M^* \le M_\lambda^*$ due to the certificate being incomplete.
> > >
> > > The authors should feel free to use other strategies/adaptions to alleviate the inconsistency. I just care about the final draft being clear regarding Def. 1 and correct regarding Thm 3.

---

> > > > ### Author Response · Authors · 2024-11-26
> > > > **Response by Authors**
> > > >
> > > > Dear Reviewer DXMv, thanks for your further comment!
> > > >
> > > > We now understand that the inconsistency arises from differing interpretations of the maximal certified perturbation size as defined in Definition 1. Specifically, we interpreted it as the solution *obtained by our method*, while you considered it in the context of a *general certified defense*. To address this, we have reverted Theorem 3 to its original formulation as presented in the initial submission.  Please let us know if this resolves your concern!

---

> ### Comment · Reviewer_DXMv · 2024-11-28
>
> Thank you for the response, my last concern is now addressed and I'm happy to support acceptance.
>
> I have one comment regarding your newly added footnote 7. You state "We leave it as future work to prove whether the derived certification is tight or not." However, the proof for Thm. 3 assumes that if a poisoned datapoint is in a relevant training set partition that the base classifier associated to that partition should be counted as "having turned". This is the reason of the $\frac{n_y-n_b+I(y<b)-1}{2}$ term in the proof, or? However, a poisoned data point may actually not be able to change the prediction of the relevant base classifier and thus, the certificate is overly pessimistic and hence, not tight, or?

---

> > ### Author Response · Authors · 2024-11-28
> >
> > Dear reviewer DXMv,
> >
> > Thanks for your comments on the tightness concern. We will simply remove the footnote 7 and change the eq.(7) in Thm.3 to “$M_\lambda ≤ M^* = ... ≤ M_\lambda^*$”. Since we are unable to update the pdf now, we promise to do so in the final version.

---

### Official Review · Reviewer_d9Hd · 2024-11-03

**Soundness:** 3
**Presentation:** 3
**Contribution:** 3
**Rating:** 6
**Confidence:** 4

**Summary:**

This paper proposes a novel certified defense framework for GNN explanations, named XGNNCert. In particular, XGNNCert generates hybrid subgraphs using subgraphs of the target graph and subgraphs of the complete graph. Based on the hybrid subgraphs, XGNNCert adopts majority voting based GNN classifier and explainer as the backbone. The authors derive a theoretical guarantee on the certified robustness of GNN classifier and explainer and provide the formula to compute the perturbation budget. Experimental evaluations on various datasets demonstrate that XGNNCert maintains the explanation accuracy and guarantees the preservation of critical explanatory edges under adversarial perturbations.

**Strengths:**

- Recent works reveal that GNN explanation is vulnerable to adversarial attacks. This paper is the first to propose a certified defense framework against the attacks.
- The certification has a deterministic guarantee, which makes XGNNCert reliable in real-world scenarios.
- The experimental results show that XGNNCert achieves desirable certification compared with other empirical defense methods.
- The paper is well-organized and the writing is easy to follow.

**Weaknesses:**

- The certification relies on the majority voting mechanism. These can lead to the global information of the target graph being corrupted. For example, in experimental results (Table 1 and Table 2), the performance of XGNNCert in FC dataset decreases consistently compared with the original GNN.
- In hybrid subgraph generation, how to ensure that the size of $\{\mathcal{E}^i\}$ is uniform? If sizes of subgraphs differ distinctly from each other, it would be better to determine the parameter $p$ more carefully, such as being dependent on each subgraph.
- XGNNCert requires to generate subgraphs of the complete graph. It can be difficult to scale due to the extra memory cost. This part seems not to be included in the complexity analysis.
- XGNNCert poses challenges on hyperparameter tuning in real-world implementation. The performance can be sensitive to some hyperparameters such as $T$, $p$, and $\gamma$, while all of them should locate in a specific range.

**Questions:**

Please see Weaknesses

---

> ### Author Response · Authors · 2024-11-21
> **Response by Authors**
>
> **We thank the reviewer for appreciating that the paper is: 1) well-organized and well-written; 2) the first certified defense for GNN explainers against the graph attacks; and 3) is reliable in real-world scenarios thanks to deterministic guarantees**
>
> **W1. Performance decrease compared with original GNN**
>
> We acknowledge that in some cases, there is a non-negligible decrease in clean accuracy due to global information corruption. However, we emphasize two key points:
> - There is an inherent tradeoff between clean accuracy and provable robustness in trustworthy (explainable) AI, as the latter accounts for the worst-case attack scenarios;
> - Besides the GNN explainer, we also provide guarantees for GNN classifiers, which makes the robustness certification more difficult.
>
> Hence, this decrease in clean accuracy is reasonable and acceptable in the context of our work. We acknowledge it is interesting future work to design certified robust XGNN with better accuracy-robustness tradeoff.
>
> **W2. How to ensure subgraph size is uniform? p dependent on each graph?**
>
> We calculate the statistics of the subgraph sizes: we first compute the *ave and std* #edges on the $T$ subgraphs of each graph, and then compute the *ave and std* #edges of all graphs. Results are shown below:
>
> | Dataset | SG+House | SG+Diamond | SG+Wheel | Benzene | FC |
> |---------------|----------|------------|----------|---------|--------|
> **Avg**   | 18.30  | 58.15  | 58.17    | 126.22  | 134.83 |
> | **Std**  | 4.70 |6.58  |8.41| 10.65| 11.29|
>
> Our observations indicate that the subgraph sizes do not vary significantly. Hence, for simplicity, we use the same $p$ for all graphs. We also emphasize that, regardless of the choice of $p$, the GNN explainer in XGNNCert identifies only the important explanatory edges in the clean graph. We will leave it as a future work to customize $p$ for individual graphs.
>
> **W3. Memory cost/scalability of XGNNCert.**
>
> Thanks for the comment! For a test graph, we generate T subgraphs with each having (p* \#nodes$^2$) edges, so the extra memory cost per graph is  O(T* p * \#nodes$^2$). Note that an edge is represented as a pair of node indexes in the implementation, which is memory efficient. In addition, we focus on the graph classification task in this paper, where the graph sizes are usually small, e.g., the average \#nodes in the used 5 benchmark graph datasets are between 10 and 22). We admit it is important future work to generalize the proposed certified defense mechanism to large-scale graphs such as social networks.
>
> *We add discussions on memory overhead in Line 482-484 in Section 4.2.4 in the revised version.*
>
> **W4. Challenges on hyperparameter tuning of XGNNCert**
>
> We acknowledge that determining the optimal hyperparameters for an AI algorithm without studying their range of values remains a long-standing and challenging problem. Thus, we conduct experiments to analyze the impact of hyperparameters on our XGNNCert and provide insights into their effects. While the performance of XGNNCert is influenced by hyperparameter values, our results demonstrate stability within specific ranges, such as $\gamma \in [0.2, 0.4]$ and $T = 50,70$. Notably, we highlight that complex AI methods all require hyperparameter tuning.

---

> > ### Author Response · Authors · 2024-11-24
> > **Official Comment by Authors**
> >
> > Dear Reviewer d9Hd,
> >
> > Thank you again for your time and effort on reviewing this paper and your constructive comments. We truly appreciate how your feedback has strengthened our paper.
> >
> > As the interaction period nears its conclusion, we would be grateful if you could confirm whether our responses meet your expectations or if more clarifications are needed within the limited time.
> >
> > Thank you once again for your valuable input.
> >
> > Best,
> >
> > Authors

---

> ### Comment · Reviewer_d9Hd · 2024-11-25
>
> I appreciate the detailed responses provided by the authors. Most of my concerns are addressed. Hence, I will keep my positive suggestion.

---

> > ### Author Response · Authors · 2024-11-25
> > **Response by Authors**
> >
> > We’re glad that all your comments have been addressed and sincerely thank you for the positive score!

---

### Comment · Area_Chair_EMra · 2024-11-24

Dear reviewers,

Thanks for serving as a reviewer. As the discussion period comes to a close and the authors have submitted their rebuttals, I kindly ask you to take a moment to review them and provide any final comments.

If you have already updated your comments, please disregard this message.

Thank you once again for your dedication to the OpenReview process.

Best,

Area Chair

---

### Meta-Review · Area_Chair_EMra · 2024-12-21

**Metareview:**

This paper proposes a novel certified defense framework for Explaining Graph Neural Network (XGNN), called XGNNCert. XGNNCert generates hybrid subgraphs by combining subgraphs from the target and complete graphs. Using these hybrid subgraphs, it employs a majority voting-based GNN classifier and explainer as the core components. The authors provide a theoretical guarantee for the certified robustness of the classifier and explainer, along with a formula for computing the perturbation budget. Experimental results on various datasets show that XGNNCert maintains explanation accuracy and ensures the preservation of critical explanatory edges under adversarial perturbations.

Strengths:

1. First certified framework for XGNNs.

2. Comprhensive evluations demonstrate the effectivenss of their method.

3. Well-written

Weaknesses:

Not esay to extend to large scale graphs currently.

Most reviewers show positive attittude towards this paper and I also believe the rest concerns have already been addressed. Therefore, I tend to accept this paper.

**Additional Comments On Reviewer Discussion:**

Most reviewers and authors dicuss about the proposed methods complexity, further evaluations, etc. And these reviewers also update their positive attitude towards this paper. Although there are still some reviewers didn't find time to respond, I believe their proposed concerns has been well addressed by the authors' rebuttal

---

### Decision · Program_Chairs · 2025-01-22

Accept (Poster)